# ONLINE CLUSTERING WITH NEARLY OPTIMAL CONSISTENCY

**T-H. Hubert Chan**
The University of Hong Kong
hubert@cs.hku.hk

**Shaofeng H.-C. Jiang**
Peking University
shaofeng.jiang@pku.edu.cn

**Tianyi Wu**
Peking University
wuty@stu.pku.edu.cn

**Mengshi Zhao**
The University of Hong Kong
zmsxsl@connect.hku.hk

## ABSTRACT

We give online algorithms for $k$-MEANS (more generally, $(k, z)$-CLUSTERING) with nearly optimal *consistency* (a notion suggested by Lattanzi & Vassilvitskii (2017)). Our result turns any $\alpha$-approximate offline algorithm for clustering into a $(1+\epsilon)\alpha^2$-competitive online algorithm for clustering with $O(k \operatorname{poly} \log n)$ consistency. This consistency bound is optimal up to $\operatorname{poly} \log(n)$ factors. Plugging in the offline algorithm that returns the exact optimal solution, we obtain the first $(1+\epsilon)$-competitive online algorithm for clustering that achieves a linear in $k$ consistency. This simultaneously improves several previous results (Lattanzi & Vassilvitskii, 2017; Fichtenberger et al., 2021). We validate the performance of our algorithm on real datasets by plugging in the practically efficient $k$-MEANS++ algorithm. Our online algorithm makes $k$-MEANS++ achieve good consistency with little overhead to the quality of solutions.

## 1 INTRODUCTION

The well-known $k$-MEANS clustering algorithm has been employed extensively in machine learning applictions. The input is a set of points $P$ in $\mathbb{R}^d$ and a parameter $k \geq 1$. The goal is to find a set of $k$ centers $C \subset \mathbb{R}^d$ such that the cost function $\operatorname{cost}(P, C) := \sum_{x \in P}(\operatorname{dist}(x, C))^2$ is minimized, where $\operatorname{dist}(x, C) := \min_{c \in C} \operatorname{dist}(x, c)$ and $\operatorname{dist}(x, c) := \|x - c\|_2$. $k$-MEANS is tightly related to $k$-MEDIAN, and the only difference is that the cost function of $k$-MEDIAN takes sum of distances without squaring.

We focus on online versions of $k$-MEANS. The online setting captures the practical scenario of evolving datasets and the requirement of making prompt decisions. In a typical online setting, data points arrive in an arbitrary order, and the algorithm must decide immediately, without knowing the entire input, whether and where to define a new center (and the data points are automatically assigned to the nearest center). As usual, the performance of an online clustering algorithm is measured by the competitive ratio, which is the ratio between the algorithm's $k$-MEANS cost and the optimal $k$-MEANS cost (with full information). The central challenge in the setting is that the decision must be made without knowing the information of the entire dataset, and the competitive ratio exactly captures this difficulty.

Unfortunately, online clustering is very sensitive to incomplete information, and it has been shown to admit strong lower bounds (Liberty et al., 2016) so that relaxations must be made to allow any finite competitive ratio. One natural relaxation is to allow recourse of decisions, and this has been formulated by Lattanzi & Vassilvitskii (2017). Specifically, let $C_i$ be the center set after the algorithm processes the $i$-th input point, then the total recourse of the algorithm is $\sum_i |C_i \setminus C_{i-1}|$. This quantity is called the *consistency* of the algorithm (Lattanzi & Vassilvitskii, 2017).

As shown by Lattanzi & Vassilvitskii (2017), there exists $O(k^2 \operatorname{poly}(\log n))$-consistent $O(1)$-competitive algorithms for $k$-MEANS and $k$-MEDIAN. They also show a lower bound that any $O(1)$-competitive algorithm must be $\Omega(k \operatorname{poly}(\log n))$-consistent. In a recent work by Fichten-

berger et al. (2021), an $O(1)$-competitive algorithm with improved consistency $O(k \operatorname{poly} \log(n))$ for $k$-MEDIAN is given. All these results also work for general metric spaces.

Despite the progress, there is still a gap in the consistency vs competitive ratio tradeoff. In particular, it is unclear if $k$-MEANS also admits $O(1)$-competitive ratio with $O(k \operatorname{poly} \log(n))$-consistency (as in $k$-MEDIAN (Fichtenberger et al., 2021)). Moreover, the lower bound in Lattanzi & Vassilvitskii (2017) does not rule out $(1+\epsilon)$ competitive ratio (although this might require an exponential running in general). Lastly, existing consistent algorithms are specially designed to optimize the ratio and might not be as efficient as practical algorithms such as $k$-MEANS++ (Arthur & Vassilvitskii, 2007), so it would be useful to turn $k$-MEANS++, or more generally any offline clustering algorithm, into an online algorithm with good consistency.

## 1.1 OUR RESULTS

Our main result, stated in Theorem 1.1, systematically addresses these challenges. Our algorithm turns any $\alpha$-approximate *offline* algorithm ($\alpha \geq 1$) into an *online* algorithm with $(1+\epsilon)\alpha^2$ competitive ratio, with little overhead in time complexity. Moreover, no matter what offline algorithm is used, it always achieves the nearly-optimal $O(k \operatorname{poly} \log n)$ consistency. Importantly, this consistency bound is nearly optimal up to $\operatorname{poly} \log n$ factors, as is complemented by a lower bound of Lattanzi & Vassilvitskii (2017).

**Theorem 1.1** (Informal; see Theorem 3.1). *Given an offline $\alpha$-approximate algorithm for $k$-MEANS that runs in $T(n)$ time, there exists an $\tilde{O}_\epsilon(k)$-consistent[1] $(1+\epsilon)\alpha^2$-competitive algorithm for online $k$-MEANS, and the running time is $\tilde{O}_\epsilon(nk + k^3 \cdot T(\tilde{O}_\epsilon(k)))$.*

We notice that the formal statement of this result more generally works for $(k, z)$-CLUSTERING (see Definition 2.1, which particularly contains $k$-MEDIAN), in addition to $k$-MEANS. Moreover, although we state our results in Euclidean $\mathbb{R}^d$ which is a typical case for clustering, our results also apply to general metrics. Our bound actually has a dependency on $\operatorname{poly}(\log \Delta)$ factor, where $\Delta$ is the aspect ratio of the dataset. This is necessary (for any constant ratio) as mentioned in Fichtenberger et al. (2021), and that one can typically assume $\Delta = \operatorname{poly}(n)$ so that $\operatorname{poly}(\log \Delta)$ translates to $\operatorname{poly}(\log n)$.

As an important corollary, if one plugs in the brute-force exact offline algorithm (which may run in exponential time), Theorem 1.1 leads to a $(1+\epsilon)$-competitive $O(k \operatorname{poly} \log n)$-consistent algorithm for $k$-MEANS (and for general $(k, z)$-CLUSTERING). This is the first $(1+\epsilon)$-competitive algorithm for any clustering problem with nontrivial consistency, while simultaneously achieving a nearly optimal consistency bound, hence fundamentally improves all relevant previous works (Lattanzi & Vassilvitskii, 2017; Fichtenberger et al., 2021) in ratio and/or consistency. Note that this $1+\epsilon$ ratio is nontrivial even allowing infinite computational power.

**Experiments.** While the mentioned $1+\epsilon$ bound is powerful, it is mostly of theoretical value because of the exponential running time. Thanks to the generality of Theorem 1.1, we are able to plug in a widely-used efficient algorithm, $k$-MEANS++ (Arthur & Vassilvitskii, 2007), to obtain an efficient consistent online clustering algorithm. We validate (in Section 5) the performance for this new consistent $k$-MEANS++ on 3 real datasets, and compare with the vanilla $k$-MEANS++ as well as a previous algorithm (Lattanzi & Vassilvitskii, 2017), as baselines. Our experiments show that with a cost similar to $k$-MEANS++ and the algorithm in Lattanzi & Vassilvitskii (2017), our algorithm achieves much lower consistency.

## 1.2 TECHNICAL OVERVIEW

At a high level, our algorithm starts with maintaining a *consistent $\epsilon$-coreset*. Roughly speaking, a coreset (Har-Peled & Mazumdar, 2004) is a small proxy of the dataset, such that for every center set, the cost on the coreset is within $1 \pm \epsilon$ to that of the original dataset. Our coreset is of size $\tilde{O}_\epsilon(k)$, and that the consistency is bounded by $\tilde{O}_\epsilon(k)$. We build such a coreset by a modified ring sampling technique (Chen, 2009; Cohen-Addad et al., 2021), and a very similar construction was also introduced in Woodruff et al. (2023). We remark that similar steps are also employed in various

---

[1] $\tilde{O}_\epsilon$ hides $\operatorname{poly}(\epsilon^{-1} \log(n))$ factor.

previous works (Lattanzi & Vassilvitskii, 2017; Fichtenberger et al., 2021), but they do not give $\epsilon$-coresets and hence it only leads to $O(1)$-approximation.

With such a small $\epsilon$-coreset, we are able to reduce the problem to the case with size-bounded input. In particular, we can work with inputs with length $\tilde{O}_\epsilon(k)$, and it suffices to have $\tilde{O}_\epsilon(k)$ total recourse. This is simpler than the original problem, since it only requires an amortized $\tilde{O}_\epsilon(1)$ recourse, instead of $o(1)$.

Our algorithm for this size-bounded input case is based on the framework of Fichtenberger et al. (2021), but our implementation differs because the original framework only works for $k$-MEDIAN and achieves only an $O(1)$ approximation ratio. Roughly speaking, the algorithm processes input points in batches (which we also call phases in our proof), using the following steps for each batch. Suppose the algorithm starts with a set of exactly $k$ centers $C_0$ as the current solution. Before processing any input point in the batch, it deletes the maximum number of centers from $C_0$ such that the cost of the resultant center set is still good enough; denoting the number of deleted centers as $\ell$ and the new center set as $C$. Then for every input point $p$, the algorithm directly include $p$ to the current center set $C$, provided that $|C|$ is still less than $k$. When $|C|$ reaches $k$ again, the algorithm "forgets" $C$ and operates on $C_0$ to conclude the batch and build the initial solution for the next batch. Specifically, it chooses $\tilde{O}_\epsilon(\ell)$ centers in $C_0$, and then swaps them with the same number of centers in the candidate center set.

Clearly, this procedure has amortized recourse $\tilde{O}_\epsilon(1)$, so the consistency follows immediately. However, obtaining the claimed ratio is nontrivial, and it particularly requires very careful choice of swapping centers. Next, we discuss how the deletion and swap is done in more detail. We focus on the case where we have the access to the optimal $k$-MEANS solutions, and the goal is to achieve $(1 + \epsilon)$-competitive. We discuss at the end how our algorithm works with a general $\alpha$-approximate (offline) algorithm.

**Procedure for deletion and swap.** For the deletion step, the $\ell$ is defined as the maximum number of centers that can be removed from $C_0$ such that the resultant center set is $(1 + O(\epsilon))$-approximate. The procedure for swap is more complicated and depends on the structure of "well separated pairs", which was also used in Fichtenberger et al. (2021) but only defined for constant approximation. Given two center sets $U$ and $V$, we define $(u, v)$ as an $\epsilon$-well separated pair for some $u \in U$ and $v \in V$, if $\mathrm{dist}(u, v)$ is $\epsilon$ times between the nearest neighbors in their respective center sets, namely, $\mathrm{dist}(v, V \setminus \{v\})$ and $\mathrm{dist}(u, U \setminus \{u\})$. In our swapping procedure, we let $U := C_0$ and $V$ be the optimal solution at the end of the phase (which is the time step when the swapping happens), and identify the points in $C_0$ that do *not* belong to a well separated pair.

Now, recall that in our algorithm outline we say the number of swapped points needs to be $\tilde{O}_\epsilon(\ell)$ in order to bound the consistency. To show this, we need our first key lemma (Lemma C.13): given a point set $P$ and two center sets $U$ and $V$, suppose $U$ and $V$ form $t$ $\epsilon$-well separated pairs, then $\Omega(k - t)$ centers can be removed from $U$ to form $U'$, such that $\mathrm{cost}(P, U') \leq \mathrm{cost}(P, U) + \epsilon(\mathrm{cost}(P, U) + \mathrm{cost}(P, V))$. Applying this key lemma with $U := C_0$ and $V$ being the optimal solution to the point set at the end of the batch, it implies that $\ell \geq \tilde{\Omega}_\epsilon(k - t)$ (since we can assume at any time step the optimal solution is within $(1 \pm \epsilon)$ factor to each other). Observe that $k - t$ is precisely the number of swaps, and this is at most $\tilde{O}_\epsilon(\ell)$.

We remark that Fichtenberger et al. (2021) proves a weaker version of this lemma whose error guarantee is only $O(\mathrm{cost}(P, U) + \mathrm{cost}(P, V))$, whereas ours is $\epsilon$ times of this. This is inherently caused by the linear programming method used in their proof. In particular, due to the integrality gap of linear programming which is a constant, this method cannot achieve a $1 + \epsilon$ ratio like ours. Furthermore, their linear programming analysis seems to only apply to $k$-MEDIAN, whereas our general technique is suitable for general $(k, z)$-CLUSTERING (in particular $k$-MEANS). Our proof cannot use linear programming, and it is inspired by the analysis of local search algorithms for the clustering problem (Cohen-Addad et al., 2016; Friggstad et al., 2019).

**Bounding the cost of swaps.** We next describe how we bound the cost of the swap step. We use a similar strategy as in Fichtenberger et al. (2021), where the key idea is to use "robust centers". Roughly speaking, a "robust center" is a center such that it is *approximately locally optimal* – any other center that is close enough to it cannot significantly improve the approximation ratio. It enables

our algorithm to build cluster centers that remain effective even as data evolves over time, without requiring knowledge of future changes. This is useful in the analysis, since in a near-optimal solution with robust centers, even moving the centers a bit does not improve the cost (significantly), which may not otherwise hold in an arbitrary approximate solution. Thus, we need to add extra steps of making centers robust in various places in the algorithm. While the robust centers possess good properties, making centers robust may introduce additional error. Nonetheless, we manage to bound the cost of the swap step in our second key lemma (Lemma C.2), based on which we inductively show if the centers are robust at the start of a batch and that the center is $(1 + O(\epsilon))$-approximate, then after the batch the (new) centers are still robust and has the same ratio. Again, similar lemmas and notions of robust centers were first introduced in Fichtenberger et al. (2021), but they only considered the $O(1)$-approximation version. Our technical contribution is to modify the definition of robust centers so that it is $(1 + \epsilon)$-approxiamte, as well as to strengthen the inductive guarantee to be $(1 + \epsilon)$ in our key lemma. The analysis uses several new steps and observations compared with that in Fichtenberger et al. (2021).

**General $\alpha$-approximation.** Now we show how the above generalizes to work with any given $\alpha$-approximate offline algorithm. Recall that $C_0$ is the center set at the beginning of a batch. In the deletion step, instead of removing maximal number of centers such that the resultant center is still a $(1 + O(\epsilon))$-approximate solution, the algorithm calls the $\alpha$-approximate offline algorithm with candidate center set as $C_0$ and $k = \ell'$ for every $1 \leq \ell' < k$. Then it picks the maximum $\ell'$ such that the cost is less than $(1 + O(\epsilon))\alpha \cot(P_0, C_0)$, where $P_0$ is the point set at the beginning of a batch. Similarly, in the swap step, we replace the optimal solution $V$ for the point set at the end of the batch with an $\alpha$-approximate one, and then perform the swaps as before (i.e., swapping out from $C_0$ points that do not belong to a well separated pair between $C_0$ and $V$). We also adopted the two key lemmas to work with these new steps.

## 1.3 RELATED WORK

Data points in the online setting that we consider is insertion-only. Recent works also study the more general dynamic setting for consistent online clustering, such that data points can also be deleted. Cohen-Addad et al. (2019) gives $O(1)$-competitive $n \operatorname{poly} \log(n)$-consistent algorithm for dynamic facility location with uniform opening cost in general metrics, and Bhattacharya et al. (2022) gives a similar bound for the more general case of non-uniform opening cost. Lacki et al. (2024) provides an $O(n)$-consistent algorithm that achieves $O(1)$ ratio for dynamic $k$-CENTER in general metrics, and they also show an $\Omega(n)$ consistency lower bound (for any finite ratio). Recently, Bhattacharya et al. (2024) provided a uniform framework for consistent $(k, z)$-CLUSTERING, achieving an $O(1/\epsilon)$ ratio $\tilde{O}(nk^\epsilon)$ consistency.

While much research focuses on optimizing resource efficiency, there are other studies aiming to minimize the number of centers in online $(k, z)$-CLUSTERING without recourse while retaining a constant approximation ratio. For example, studies have explored scenarios such as (i) using an estimate of the optimal clustering cost as an input parameter, (ii) knowing the total number of points $n$ in advance, or (iii) processing points in random order Moshkovitz (2021); Bhaskara & Ruwanpathirana (2020); Bhattacharjee & Moshkovitz (2021); Hess et al. (2021). These studies, along with many online clustering algorithms, do not allow for recourse and require opening more than $\Omega(k \log(n))$ centers, leading to a bi-criteria approximation.

## 2 PRELIMINARIES

Although we focus on $\mathbb{R}^d$, we assume the dataset is a subset of $[\Delta]^d$ for some integer $\Delta$, which is supposed to be the aspect ratio. This is without loss of generality since one can always rescale the dataset provided the knowledge of $\Delta$. Let $\mathcal{C} \subseteq \mathbb{R}^d$ be the candidate center set. Given a point $q \in \mathbb{R}^d$ and a radius $r \in \mathbb{R}_+$, a ball is defined as $\operatorname{ball}(q, r) = \{p \in \mathbb{R}^d \mid \operatorname{dist}(p, q) \leq r\}$. Moreover, a ball with respect to a point set $P$ is defined as $\operatorname{ball}_P(q, r) := \operatorname{ball}(q, r) \cap P$.

**Definition 2.1** ($(k, z)$-CLUSTERING). Given $P \subseteq \mathbb{R}^d$, for any center set $C \subseteq \mathcal{C}$ with $|C| \leq k$, define the cost for $(k, z)$-CLUSTERING as

$$\operatorname{cost}_z(P, C) := \sum_{x \in P} (\operatorname{dist}(x, C))^z.$$

We define $\mathrm{OPT}(P)$ as the optimal cost of $(k, z)$-CLUSTERING. Specifically,

$$\mathrm{OPT}_z(P) = \min_{C \subseteq \mathcal{C}, |C| \leq k} \mathrm{cost}_z(P, C).$$

**Definition 2.2** (Online $(k, z)$-CLUSTERING). In online $(k, z)$-CLUSTERING, the input dataset $P$ is presented as a sequence $(p_1, \ldots, p_n)$, and upon the arrival of $p_i$ (for every $i \in [n]$), the online algorithm must output a center set $C_i \subseteq \mathcal{C}$ such that $|C_i| \leq k$. We say an online algorithm is $\rho$-competitive if $\mathrm{cost}_z(P, C_i) \leq \rho \cdot \mathrm{OPT}_z(P_i)$ for every $i$, where $P_i$ is the first $i$ points in $P$.

**Definition 2.3** (Consistency). We say a sequence of center sets $C_1, \ldots, C_n$ $\Gamma$-consistent, if $\sum_{t \in [n]} |C_t \setminus C_{t-1}| \leq \Gamma$ ($C_0 := \emptyset$). An online algorithm is $\Gamma$-consistent if its output center sets $C_1, \ldots, C_n$ is $\Gamma$-consistent.

**Weighted point set.** A weighted point set is an (ordinary) point set equipped with a positive weight function. Specifically, we identify a weighted point by a vector $\vec{P} \in \mathbb{R}^{[\Delta]^d}$ with support $P$. Then for $\vec{P}$, the weight of a point $p \in P$ is denoted as $\vec{P}(p)$, and for a subset $S \subseteq P$, we define $\vec{P}(S) := \sum_{p \in S} \vec{P}(p)$. We also use $|\vec{P}|$ to represent $\vec{P}(P)$ as the total weight of the point set. We treat unweighted point sets as weighted point sets with unit weight.

The relation $\vec{P}_1 \subseteq \vec{P}_2$ is defined as $\vec{P}_1 \leq \vec{P}_2$ with respect to coordinate-wise comparison. The operation $\vec{P}_1 \cup \vec{P}_2$ is the coordinate-wise maximum of the two vectors, and $\vec{P}_1 \cap \vec{P}_2$ is the coordinate-wise minimum. The set subtraction $\vec{P}_1 \setminus \vec{P}_2$ sets every negative value in $\vec{P}_1 - \vec{P}_2$ to zero. The cost function for $(k, z)$-CLUSTERING on weighted point set is generalized as $\mathrm{cost}_z(\vec{P}, C) := \sum_{p \in P} \vec{P}(p)(\mathrm{dist}(p, C))^z$. The ball with respect to a weighted point set is generalized as $\mathrm{ball}_{\vec{P}}(q, r) := \{\vec{p} \in \vec{P} \mid \mathrm{dist}(p, q) \leq r\}$.

**Definition 2.4.** For $0 < \epsilon < 1$ and weighted set $\vec{P}$, a weighted set $\vec{S}$ such that $S \subseteq P$ is an $\epsilon$-coreset for $(k, z)$-CLUSTERING if

$$\forall C \subseteq \mathcal{C}, |C| \leq k, \quad (1 - \epsilon) \mathrm{cost}_z(\vec{P}, C) \leq \mathrm{cost}_z(\vec{S}, C) \leq (1 + \epsilon) \mathrm{cost}_z(\vec{P}, C).$$

**Definition 2.5** (Cluster of a point). Let $\vec{P}$ be a weighted point set, $C \subseteq \mathcal{C}$ be a center set and $c \in C$ be a center. The cluster of $c$ with respect to $\vec{P}$ is defined as (where ties are broken arbitrarily and consistently)

$$\vec{P}[C, c] := \{\vec{p} \in \vec{P} \mid \mathrm{dist}(p, C) = \mathrm{dist}(p, c)\}.$$

**Lemma 2.6** ([Cohen-Addad et al. (2021)](#)). *Let $a, b, c$ be any points in $[\Delta]^d$, $z$ be any positive constant. Then for any $\epsilon > 0$,*

$$\mathrm{dist}(a, b)^z \leq (1 + \epsilon)^{z-1} \mathrm{dist}(a, c)^z + \left(\frac{1 + \epsilon}{\epsilon}\right)^{z-1} \mathrm{dist}(b, c)^z,$$

$$|\mathrm{dist}(a, c)^z - \mathrm{dist}(b, c)^z| \leq \epsilon \cdot \mathrm{dist}(a, c)^z + \left(\frac{2z + \epsilon}{\epsilon}\right)^{z-1} \mathrm{dist}(a, b)^z.$$

## 3 FRAMEWORK AND PROOF OF MAIN THEOREM

In this section, we present the high-level framework and explain how it proves Theorem 3.1 which is our main theorem. All results stated in this section are without the actual dependence in $z$, since the most interesting case is $z \in \{1, 2\}$ anyway. However, we do try to figure out the detailed dependence in the proof (Section 4).

**Theorem 3.1.** *Suppose for some $\alpha \geq 1$ there is an $\alpha$-approximate (randomized) offline algorithm for $(k, z)$-CLUSTERING that runs in $T(n)$ time. Then there exists an algorithm for online $(k, z)$-CLUSTERING such that for every $n$-point dataset in $[\Delta]^d$ ($\Delta \geq 1$ is integer), $0 < \epsilon < 1$ and integers $k, z \geq 1$, it is $(1 + \epsilon)\alpha^2$-competitive and $O(\alpha dk \, \mathrm{poly}(\epsilon^{-z} \log(n\Delta)))$-consistent[2] with probability $1 - \frac{1}{\mathrm{poly}(n)}$, provided that all invocations of the offline algorithm succeed. The algorithm runs in $O(kn \log(n\Delta) + 2^{\mathrm{poly}(\epsilon^{-1})} dk^3 \, \mathrm{poly}(\epsilon^{-1} \log(n\Delta)) T(dk \, \mathrm{poly}(\epsilon^{-1} \log(n\Delta))))$ time.*

---

[2] Notice that the factor $d$ may be turned into $O(\log n)$ by using a dimension reduction ([Makarychev et al., 2019](#)).

Our proof idea consists of two steps. First, we reduce the number of points to be considered by leveraging a coreset structure, which contains significantly fewer points than the original set, with each point assigned a positive weight. This coreset structure satisfies two key properties: it is always a coreset for any prefix of the stream, and points sampled into the data structure are never deleted from the incremental coreset. We note that with slight modifications, the so-called "online coreset" proposed in Woodruff et al. (2023) meets our requirements. Although their definition of an online coreset seems to require the algorithm to access the entire dataset, their main procedure can indeed be executed in an online/incremental manner. We summarize our coreset guarantee in the following lemma.

**Lemma 3.2** (Consistent coreset (Woodruff et al., 2023)). *There exists an algorithm that given as input $0 < \epsilon < 1$ integers $k, z \geq 1$ and an online point sequence $(p_1, \ldots, p_n)$ in $[\Delta]^d$, outputs a weighted point set sequence $\vec{D}_1, \ldots, \vec{D}_t$ such that*

1. *There is $m := O(\log(n\Delta))$ numbers $e_1 := 1 \leq \ldots \leq e_m := n + 1$ such that for every $1 \leq i \leq m-1$, $\mathrm{OPT}_z(\vec{D}_{e_{i+1}-1}) \leq 2\mathrm{OPT}_z(\vec{D}_{e_i})$ and for every $e_i \leq j \leq e_{i+1}-1$, $\vec{D}_j \subseteq \vec{D}_{j+1}$.*
2. *With probability $1 - \frac{1}{\mathrm{poly}(n)}$, for all $1 \leq i \leq n$, $\vec{D}_i$ is an $\epsilon$-coreset for $P_i := \{p_1, \ldots, p_i\}$ and $|\vec{D}_i| = O(dk\,\mathrm{poly}(\epsilon^{-1}\log(n\Delta)))$.*

*The algorithm runs in $O(kn\log(n\Delta))$ time.*

In the second step, we solve the consistent $(k, z)$-CLUSTERING with weighted points as the input. Specifically, our algorithm (in Lemma 3.3) achieves $\tilde{O}(1)$ amortized consistency. The proof of this lemma is our main technical contribution and is postponed to Section 4.

**Lemma 3.3** (Algorithm for bounded input). *Suppose for some $\alpha \geq 1$ there is an $\alpha$-approximate (randomized) offline algorithm for $(k, z)$-CLUSTERING that runs in $T(n)$ time. Then there exists an algorithm for online $(k, z)$-CLUSTERING such that for every weighted point set $\vec{P}_0$ and weighted $m$-point sequence $(\vec{p}_1, \ldots, \vec{p}_m)$ in $[\Delta]^d$, provided that $\mathrm{OPT}_z(\vec{P}_0 \cup \{\vec{p}_1, \ldots, \vec{p}_m\}) \leq 2\mathrm{OPT}_z(\vec{P}_0)$, and all invocations of the offline algorithm succeed, it is $(1 + \epsilon)\alpha^2$-competitive and $O(\alpha m\,\mathrm{poly}(\epsilon^{-1}\log(\Delta)))$-consistent. The algorithm runs in $O(mk^2 T(m) + dkm^2 2^{\mathrm{poly}(\epsilon^{-1})}\log(\Delta))$ time.*

Finally, we combine these two steps to form Algorithm 1. The bound on cost and consistency follows from the composition of Lemma 3.2 and Lemma 3.3. The formal proof of Theorem 3.1 is presented in Section A.

---

**Algorithm 1:** Consistent online algorithm for $(k, z)$-CLUSTERING

1 denote the algorithms in Lemma 3.2 and Lemma 3.3 as $\mathcal{A}^{\mathrm{coreset}}, \mathcal{A}_\alpha^{\mathrm{bound}}$, respectively
2 **for** $i = 1, \ldots, n$, *suppose $x_i$ is inserted* **do**
3     feed $x_i$ to $\mathcal{A}^{\mathrm{coreset}}$ and let $D_i$ be the coreset returned by $\mathcal{A}^{\mathrm{coreset}}$
4     **if** $i \leq k$ **then**
5         $C_i \leftarrow C_{i-1} \cup \{x_i\}$ /* let $C_0 \leftarrow \emptyset$                                           */
6     **else**
7         **if** $\vec{D}_{i-1} \subseteq \vec{D}_i$ **then**
8             feed $\vec{D}_i \setminus \vec{D}_{i-1}$ to $\mathcal{A}_\alpha^{\mathrm{bound}}$
9         **else**
10             start $\mathcal{A}_\alpha^{\mathrm{bound}}$ over and feed it $\vec{D}_i$
11         let $C_i$ be the output of $\mathcal{A}_\alpha^{\mathrm{bound}}$
12 **return** $C_1, \ldots, C_n$

---

## 4 ALGORITHMS FOR BOUNDED INPUT: PROOF OF LEMMA 3.3

In this section, we give the algorithm for inputs of bounded length, denoting the length as $m$. Our algorithm is $(1 + \epsilon)\alpha^2$-competitive with consistency proportional to $m$, which proves Lemma 3.3.

We start with several definitions used in our algorithm in Section 4.1. This includes a key notion called "robust sequence" in Section 4.1, where a similar version has been proposed in Fichtenberger et al. (2021). Moreover, a procedure for identifying a robust sequence (in Algorithm 3) is crucially used in our algorithm. We state our main algorithm in Section 4.2. The analysis of the algorithm can be found in Section C, and combining these immediately conclude Lemma 3.3.

## 4.1 ROBUST (CENTER) SEQUENCES

In this section (and in Section B), we give the definition of robust sequence in Definition 4.1 along with many other related properties and definitions. All these essentially follow similar ones in Fichtenberger et al. (2021), whereas the key difference is that ours works for $(1 + \epsilon)$-approximation and general $(k, z)$-CLUSTERING. Because of this, we need to re-prove all properties so that they comply with our stronger definition.

**Definition 4.1** (Robust (center) sequence). Let $t \geq 1$ and $\vec{P}$ be a weighted point set. A robust center sequence $(c_0, c_1 \ldots, c_t)$ can be constructed by the following procedure. Start with any center $c_t$. For $i$ from $t$ to 1, we pick $c_{i-1}$ as follows.

1. $c_{i-1} \leftarrow c_i$, when at least one of the following holds:
   (i) $\mathrm{avgcost}_z(\mathrm{ball}_{\vec{P}}(c_i, (1 + \epsilon)^{\frac{i}{z}}), c_i) \geq \frac{\epsilon^{2z}}{9^z z^z}(1 + \epsilon)^i$; or,
   (ii) $\mathrm{avgcost}_z(\mathrm{ball}_{\vec{P}}(c_i, (1 + \epsilon)^{\frac{i}{z}}), c_i) \leq (1 + \epsilon) \min_{c' \in \mathcal{C}} \mathrm{avgcost}_z(\mathrm{ball}_{\vec{P}}(c_i, (1 + \epsilon)^{\frac{i}{z}}), c')$.
2. Otherwise, $c_{i-1}$ is any center $c \in \mathcal{C}$ such that:
   $\mathrm{avgcost}_z(\mathrm{ball}_{\vec{P}}(c_i, (1 + \epsilon)^{\frac{i}{z}}), c) \leq (1 + \epsilon) \min_{c' \in \mathcal{C}} \mathrm{avgcost}_z(\mathrm{ball}_{\vec{P}}(c_i, (1 + \epsilon)^{\frac{i}{z}}), c')$

Given a sequence $(c_0, c_1, c_2, \ldots, c_t)$ and $1 \leq t_0 \leq t$, the $t_0$-prefix of the sequence is defined as the prefix $(c_0, c_1, \ldots, c_{t_0})$. Given a center sequence with length $t + 1$, it is called $t_0$-prefix robust if $t_0 \leq t$ and its $t_0$-prefix is robust. It is immediate that (i) any prefix of a robust center sequence is automatically robust and (ii) given a number $t_0$ and a sequence $(c_0, c_1, \ldots, c_t)$, the sequence is not $t_0$-prefix robust if either $t_0 > t$ or the prefix $(c_0, c_1, \ldots, c_{t_0})$ is not robust.

The concept of a robust center sequence is pivotal in ensuring stability and adaptability in clustering, particularly when dealing with dynamic or evolving data. Intuitively, this sequence starts from some arbitrary center $c_t$, then it identifies for every distance scale (up to some threshold) a good center to replace $c$. From this point, the definition can be viewed as a backward induction algorithm with input $t$ and an initial center $c_t$. The output is the sequence $(c_0, c_1, \ldots, c_t)$ computed by the definition.

From a high-level point, a robust center sequence embodies local optimality across multiple scales, ensuring each center is resilient to minor cluster perturbations. This backward construction ensures that $c_0$ is robust at the finest scale, having been refined through progressively larger neighborhoods. This definition enforces that each center is either inherently stable or replaced by a locally near-optimal alternative, balancing current performance with adaptability to future changes. By iterating over exponentially growing radii, the sequence accounts for uncertainty in cluster scales, as clusters may develop, merge, or split over time.

A similar version of the following definition is also originally proposed in Fichtenberger et al. (2021), and we generalize it to achieve $(1 + \epsilon)$-approximation as well as to handle general $(k, z)$-CLUSTERING.

**Definition 4.2** (Bounded robust). A *witness* of a center $c \in \mathcal{C}$ is defined as a finite sequence of points in $\mathcal{C}$ that starts at $c$. Let $C \subseteq \mathcal{C}$ be a center set. A witness mapping $\mathrm{wit}_C$ maps each $c \in C$ to a witness $\mathrm{wit}_C(c)$. Given a weighted point set $\vec{P}$, $(C, \mathrm{wit}_C)$ is called *bounded robust* w.r.t. $\vec{P}$ if for every $c \in C$, $\mathrm{wit}_C(c)$ is $t_C(c)$-prefix robust with respect to $\vec{P}$, where $t_C(c)$ denotes the smallest integer $t'$ such that $(1 + \epsilon)^{\frac{t'}{z}} \geq \mathrm{dist}(c, C \setminus \{c\})/10$. If $C \setminus \{c\} = \emptyset$, we define $\mathrm{dist}(c, \emptyset) = \sqrt{d}\Delta$ (which is the largest possible distance of the data).

We define an algorithm MAKEROBUST (Algorithm 3) to make a center set with a witness mapping bounded robust, by iteratively calling a subroutine in Algorithm 2. In Algorithm 2, if $\mathrm{wit}_C(c)$ is not $t_C(c)$-prefix robust, then ROBUSTIFY$(\vec{P}, C, c)$ generates a new center point equipped with a witness that is $t_C(c)$-prefix robust. Again, these two procedures are also based on similar ones in Fichtenberger et al. (2021) but our version is generalized to work for $(1 + \epsilon)$-approximation.

---

**Algorithm 2:** Robustify, on weighted point set $\vec{P}$, center set $C$ and a center point $c \in C$

---

**1** let $t$ be the smallest integer such that $(1 + \epsilon)^{\frac{t}{z}} \geq \mathrm{dist}(c, C \setminus \{c\})/5$
**2** compute a robust sequence $(c_0, c_1, \ldots, c_t)$ such that $c_t = c$ by Definition 4.1
**3** $v \leftarrow (c_0, c_1, \ldots, c_t)$
**4** **return** $(c_0, v)$

---

**Algorithm 3:** MakeRobust, on weighted point set $\vec{P}$ and center set $C$ with witness mapping $\mathrm{wit}_C$

---

**1** **while** *There is a center $c \in C$ such that $\mathrm{wit}_C(c)$ is not $t_C(c)$-prefix robust with respect to $\vec{P}$* **do**
**2** $\quad$ Arbitrarily pick a center $c$ such that the while condition holds
**3** $\quad$ $(c_0, v) \leftarrow \mathrm{ROBUSTIFY}(\vec{P}, C, c)$
**4** $\quad$ $C \leftarrow C \setminus \{c\} \cup \{c_0\}$, $\mathrm{wit}_C(c_0) \leftarrow v$
**5** **return** $(C, \mathrm{wit}_C)$

---

### 4.2 CONSISTENT CLUSTERING ALGORITHM

Now we are ready to introduce the consistent algorithm. The algorithm runs in phases. The input of each phase is a weighted point set $\vec{P}_0$ and a center set with witness mapping $(U_0, \mathrm{wit}_{U_0})$. The algorithm guarantees that $(U_0, \mathrm{wit}_{U_0})$ is bounded robust (see Definition 4.2) with respect to $\vec{P}_0$ and $U_0$ is a $(1 + 17\epsilon)\alpha$-approximate solution for $\vec{P}_0$. Our analysis mostly focuses on an (arbitrarily) fixed phase. Let $\vec{P}_i$ be the point set after the $i$-th insertion of the phase. The algorithm executes the following steps in each phase.

1. **Deleting centers**. The algorithm enumerates $\ell \in [k]$ and runs the $\alpha$-approximate algorithm for $(k - \ell)$-clustering with point set $\vec{P}_0$ and candidate center set $U_0$. Then it picks the maximum $\ell_0$ such that the cost output by the $\alpha$-approximate algorithm is less than $(1 + 12\epsilon)\alpha \, \mathrm{cost}_z(\vec{P}_0, U_0)$. Let $\bar{U}$ be the center set with $k - \ell_0$ centers that the $\alpha$-approximate algorithm outputs.
2. **Handling insertions**. As long as the center set $\bar{U}$ consists of less than $k$ points, if a point is inserted at a position with no center (in $\bar{U}$), the algorithm includes the new point in $\bar{U}$. Otherwise, it does nothing. The center set $\bar{U}$ is the reported center set of the online algorithm.
3. **Swapping centers**. Suppose after some $\ell \geq \ell_0$ insertions from the start of the phase, the size of the center set $\bar{U}$ achieves $k$. Now, for the next input point (which is the $\ell + 1$-th insertion from the start), the algorithm starts with changing $O(\frac{2^{O(z \log(z))}\ell}{\epsilon^{8z-3}})$ centers in $U_0$ to produce a $(1 + 5\epsilon)\alpha$-approximate solution $W$ for $\vec{P}_{\ell+1}$, whose detail is given immediately after the description of the algorithm. Let $\mathrm{wit}_W(c) = (c, c)$ for every $c \in W \setminus U_0$. (This $W$ is *not* the output of the $\ell + 1$-th round yet.)
4. **Robustifying centers**. Let $(U_{\ell+1}, \mathrm{wit}_{U_{\ell+1}}) := \mathrm{MAKEROBUST}(\vec{P}_{\ell+1}, (W, \mathrm{wit}_W))$ (defined in Algorithm 3) and it is both the output of the $\ell + 1$-th insertion and the input for the next phase. Define the length of the phase as $\ell + 1$.

The detailed implementation of step 3 requires the following notion of $\epsilon$-well separated pairs, which identifies close center pairs from two different center sets. In general, this is a standard notion in geometric approximation algorithms (see Har-Peled (2011)), and similar notions have been used in local search algorithms for clustering Friggstad et al. (2019). Fichtenberger et al. (2021) also employs this notion, but only for $O(1)$-approximation whereas ours can work for $(1+\epsilon)$-approximation.

**Definition 4.3** (Well separated pairs)**.** Suppose $U$ and $V$ are two points sets, a pair $(u, v) \in U \times V$ is a $\epsilon$-well separated pair if both of the following hold:

$$\mathrm{dist}(u, v) \leq \epsilon \, \mathrm{dist}(U \setminus \{u\}, u), \quad \mathrm{dist}(u, v) \leq \epsilon \, \mathrm{dist}(V \setminus \{v\}, v) \tag{1}$$

**Implementation details of step 3.** We formalize the step 3 as follows. At the $(\ell + 1)$-st input point from the start of the phase, the algorithm runs the offline $\alpha$-approximate algorithm for the

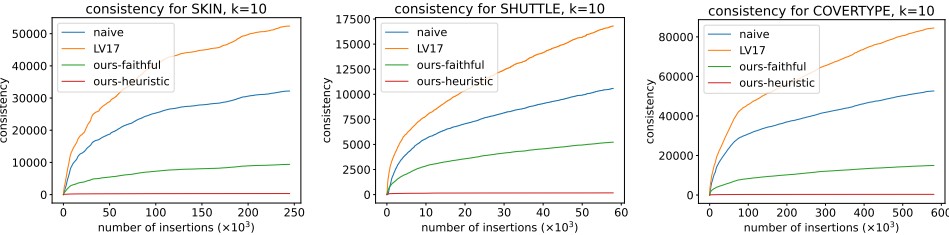

Figure 1: The consistency curve over the insertions of points, for all datasets and $k = 10$.

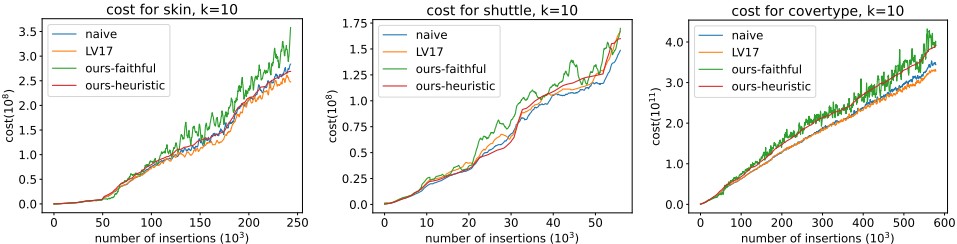

Figure 2: The cost curve over the insertions of points, for all datasets and $k = 10$. We plot the curve after applying a moving average with a window size equal to 1% of the dataset size.

current point set, denote the approximate solution as $V$. Let $s$ be some integer such that for every $i \in [s]$, $(u_i, v_i)$ forms a $\frac{\epsilon^4}{200z}$-well separated pair, and $\vec{P}_{\ell+1}[V, v_i] \subseteq \vec{P}_0$. Then the algorithm swaps centers $v_{s+1}, \ldots, v_k$ into $U_0$, forming a new center set $W := \{u_0, \ldots, u_s, v_{s+1}, \ldots, v_k\}$. Let $\text{wit}_W(c) = (c)$ for every $c \in W \setminus U_0$ and $\text{wit}_W(c) = \text{wit}_{U_0}(c)$ for others. We would relate this $s$ with $\ell$ in Lemma C.8, and it would be useful for bounding the consistency.

This finishes the description of our main algorithm. To finish the proof of Lemma 3.3, it remains to show that the algorithm has the claimed time complexity, competitive ratio and consistency. We prove these in Lemmas C.1, C.3 and C.6, to be presented in Section C.

## 5 EXPERIMENTS

In this section, we present experimental results evaluating the performance of our algorithms. We provide two implementations: one that faithfully implements Algorithm 1 and a heuristic variant (implementation details of which are described shortly). We evaluate both the computational cost and consistency of our two implementations against two baseline approaches using three real-world datasets.

**Datasets.** Our experiment is conducted on the SKIN (Bhatt & Dhall, 2009), SHUTTLE (Catlett), and COVERTYPE (Blackard, 1998) datasets from the publicly available UCI repository, which were also used in the experiments of previous works such as (Lattanzi & Vassilvitskii, 2017). The SKIN dataset consists of 245,057 points with 3 features, where each point represents an RGB pixel. The SHUTTLE dataset contains 58,000 points with 7 features. The COVERTYPE dataset consists of 581,012 points with 54 features.

**Implementation details** Now we detail our two implementations of the algorithm, both of which share a common online coreset construction phase followed by different downstream clustering approaches.

- Common Component: Online Coreset Construction. Both implementations begin with the construction of an online coreset. For this step, rather than using the worst-case coreset size bound specified in Lemma 3.2, we directly set a target size. Our experiments employ coreset sizes ranging from 1000 to 2000 elements, calibrated according to the dataset size. As our subsequent

experimental results demonstrate, these sizes provide sufficient accuracy while maintaining computational efficiency.

- Implementation 1: Consistent $k$-MEANS++. Our first implementation faithfully executes the downstream component of Algorithm 1. This algorithm incorporates a subroutine that implements a consistent clustering approach for bounded input, which requires an offline clustering algorithm. In our experiments, we integrate $k$-MEANS++ (Arthur & Vassilvitskii, 2007) into this framework. This implementation performs the complete "robustify-delete-swap" process as specified in the theoretical algorithm. We refer to this implementation as "ours-faithful" throughout our experimental evaluation.

- Implementation 2: Heuristic Single-Swap Approach. Our second implementation employs a simplified heuristic for the downstream component while using the same coreset construction algorithm as Implementation 1. This heuristic employs a simple single-swap algorithm that operates as follows: whenever a new point is added to the coreset, the algorithm evaluates whether replacing any current center point with this new point would reduce the clustering cost. If such cost-reducing swaps exist, the algorithm executes only the single swap that yields the greatest improvement. The primary advantage of the single swap approach is its small consistency and fast running time. Specifically, given a sequence $(x_1, \ldots, x_m)$ as the bounded-size input, the single swap algorithm has consistency at most $m$ and achieves $O(km)$ running time. We refer to this heuristic implementation as "ours-heuristic" in our experiments.

**Baselines.** We compare our two implementations with two baseline algorithms. Note that both of our two implementations are combining the consistent coreset (Lemma 3.2) and a downstream consistent algorithm for bounded input (Lemma 3.3 and single swap algorithm). Hence, a natural naive baseline is directly running $k$-MEANS++ on the consistent coreset instead of running any additional algorithm. Specifically, whenever the consistent coreset updates, the algorithm computes a new center set, and we note that this algorithm is $\tilde{O}(k^2)$-consistant in the worst case. Another baseline algorithm is the consistent $k$-MEANS algorithm proposed in Lattanzi & Vassilvitskii (2017)[3], which is the state-of-the-art for consistent $k$-MEANS before our work. This algorithm is constant competitive and $O(k^2 \log(n\Delta))$-consistent. These two baselines are called "naive" and "LV17", respectively.

**Experiment results.** We depict the consistency and cost curves in Figures 1 and 2 for $k = 10$, comparing the performance of the algorithms "naive", "LV17", "ours-faithful", and "ours-heuristic", as described earlier. Interestingly, the heuristic algorithm "ours-heuristic" outperforms all other algorithms in both cost and consistency, particularly hundreds times better in consistency. The strong performance of our heuristic may be attributed to the characteristic of the datasets used in our evaluation. However, it is important to note that our heuristic approach lacks worst-case approximation guarantees, so it may perform poorly on more complex datasets, for example, datasets with overlapping groups, significant outliers, or highly imbalanced cluster sizes—would likely challenge our heuristic approach more severely.

For algorithm "ours-faithful", from Figure 1, it achieves much better consistency compared to baselines, with 3 - 5 times better to naive and roughly 2 times better than Lattanzi & Vassilvitskii (2017). Moreover, achieving such a better consistency bound does not incur must overhead to the cost, as can be seen from Figure 2.

Finally, we observe that the cost has some sudden fluctuations in Figure 2. This is mainly caused by the consistent coreset, on which all baselines and our algorithm are based, since the coreset needs to recompute from scratch for $O(\text{poly}(\log n))$ times, and each recomputation introduces a big difference in cost. In addition, algorithm "ours-faithful" further uses the subroutine in Lemma 3.3, which introduces another source of re-computation during phase transition (and hence one can see an even bigger "spike" in this algorithm). Luckily, these fluctuations are not significant and are averaged out over the entire run.

---

[3]The algorithm in Lattanzi & Vassilvitskii (2017) is also a generic one like ours, and we plug in $k$-MEANS++ as we do for our algorithm.

ACKNOWLEDGMENT

Shaofeng H.-C. Jiang was supported in part by a national key R&D program of China No. 2021YFA1000900 and a startup fund from Peking University. T-H. Hubert Chan was partially supported by the Hong Kong RGC grant 17203122.

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

# Appendices

## A    PROOF OF THEOREM 3.1

**Theorem 3.1.** *Suppose for some $\alpha \geq 1$ there is an $\alpha$-approximate (randomized) offline algorithm for $(k, z)$-CLUSTERING that runs in $T(n)$ time. Then there exists an algorithm for online $(k, z)$-CLUSTERING such that for every $n$-point dataset in $[\Delta]^d$ ($\Delta \geq 1$ is integer), $0 < \epsilon < 1$ and integers $k, z \geq 1$, it is $(1 + \epsilon)\alpha^2$-competitive and $O(\alpha dk \operatorname{poly}(\epsilon^{-z} \log(n\Delta)))$-consistent[4] with probability $1 - \frac{1}{\operatorname{poly}(n)}$, provided that all invocations of the offline algorithm succeed. The algorithm runs in $O(kn \log(n\Delta) + 2^{\operatorname{poly}(\epsilon^{-1})} dk^3 \operatorname{poly}(\epsilon^{-1} \log(n\Delta)) T(dk \operatorname{poly}(\epsilon^{-1} \log(n\Delta))))$ time.*

*Proof of Theorem 3.1.* Equipped with Lemmas 3.2 and 3.3, our main algorithm is listed in Algorithm 1. By Lemma 3.2, $\mathcal{A}^{\text{coreset}}$ partitions the point stream $(p_1, p_2, \ldots, p_n)$ into $m := O(\log(n\Delta))$ parts. We use $e_1 := 1, e_2, \ldots, e_m := n + 1$ to represent the times such that $\vec{D}_{e_i - 1} \not\subseteq \vec{D}_{e_i}$. Now for a fixed part $(p_{e_i}, p_{e_i+1}, \ldots, p_{e_{i+1}-1})$, we have $|\vec{D}_{e_{i+1}-1} \setminus \vec{D}_{e_i}| \leq O(dk \operatorname{poly}(\epsilon^{-1} \log(n\Delta)))$.

Let $\ell_{e_i} := |\vec{D}_{e_{i+1}-1} \setminus \vec{D}_{e_i}|$. Note that for every $1 \leq i < m$ we have $\operatorname{OPT}(\vec{D}_{e_{i+1}-1}) \leq 2 \operatorname{OPT}(\vec{D}_{e_i})$. So feed $\vec{D}_{e_i}$ as $P_0$ and $\vec{D}_{e_{i+1}-1} \setminus \vec{D}_{e_i}$ as the point stream to $\mathcal{A}_{\alpha}^{\text{bound}}$, we get a center set sequence $(C_0^{(e_i)}, C_1^{(e_i)}, \ldots, C_{\ell_{e_i}-1}^{(e_i)})$ such that for each $0 \leq j \leq e_{i+1} - 1$, $C_j^{(e_i)}$ is a $(1 + \epsilon)\alpha^2$-approximate solution for point set $\vec{D}_{e_i+j}$ and thus a $(1 + 3\epsilon)\alpha^2$-approximate solution for $P_{e_i+j}$. Since this is true for all $e_i$, we finish the proof of the approximation ratio.

For the consistency, we directly compute it as

$$\sum_{i=1}^{m} \left[ \sum_{j=1}^{\ell_{e_i}-1} |C_j^{(e_i)} \setminus C_{j-1}^{(e_i)}| + |C_{\ell_{e_i}-1}^{(e_i)} \setminus C_0^{(e_{i+1})}| \right] \leq \sum_{i=1}^{m} \left[ O(\alpha \ell_{e_i} \operatorname{poly}(\epsilon^{-1} \log(\Delta))) + k \right]$$

$$\leq O(\alpha \operatorname{poly}(\epsilon^{-1} \log(\Delta)) \sum_{i=1}^{m} \ell_{e_i}) + km = O(\alpha dk \operatorname{poly}(\epsilon^{-1} \log(n\Delta))).$$

Finally, provided that all invocations of the offline algorithm succeed, the randomness only comes from the construction of the coreset. Thus, the failure probability inherits from Lemma 3.2, which is $\frac{1}{\operatorname{poly}(n)}$.

For the running time, note that constructing the consistency coreset takes $O(kn \log(n\Delta))$ time. Also, observing that each time we feed $\mathcal{A}_{\alpha}^{\text{bound}}$ at most $2^{-O(z)} k \operatorname{poly}(\epsilon^{-z} \log(n\Delta))$ points, Summing over all restarts of $\mathcal{A}_{\alpha}^{\text{bound}}$, the time complexity is $O(kn \log(n\Delta) + 2^{\operatorname{poly}(\epsilon^{-1})} dk^3 \operatorname{poly}(\epsilon^{-1} \log(n\Delta)) T(dk \operatorname{poly}(\epsilon^{-1} \log(n\Delta))))$. Thus we finish the proof.    □

## B    PROPERTIES OF ROBUST (CENTER) SEQUENCES

**Fact B.1.** *If a sequence $(c_0, c_1, \ldots, c_t)$ is robust, then it is $t_0$-prefix robust for every $0 \leq t_0 \leq t$.*

**Fact B.2.** *Given a point set $\vec{P}$, a robust sequence $(c_0, \ldots, c_t)$ with respect to $\vec{P}$ and an integer $1 \leq i \leq t$, for every $j < i$, $\operatorname{avgcost}_z(\operatorname{ball}_{\vec{P}}(c_j, (1 + \epsilon)^{\frac{i}{z}}), c_j) \leq \operatorname{avgcost}_z(\operatorname{ball}_{\vec{P}}(c_j, (1 + \epsilon)^{\frac{i}{z}}), c_i)$.*

The properties of Definition 4.1 are mostly used in Lemma B.3. In other words, Definition 4.1 is not directly used in most of other parts of the proof, instead, they use Lemma B.3.

**Lemma B.3.** *If $(c_0, c_1, \ldots, c_t)$ is robust, then*

- *For every $1 \leq i \leq t$, $\operatorname{dist}(c_{i-1}, c_i) \leq \frac{2\epsilon^2}{9z}(1 + \epsilon)^{\frac{i}{z}}$.*
- *For every $1 \leq i \leq t$, $\operatorname{ball}_{\vec{P}}(c_{i-1}, (1 + \epsilon)^{\frac{i-1}{z}}) \subseteq \operatorname{ball}_{\vec{P}}(c_i, (1 + \epsilon)^{\frac{i}{z}})$.*

---

[4]Notice that the factor $d$ may be turned into $O(\log n)$ by using a dimension reduction (Makarychev et al., 2019).

- $\forall i \in [n]$, $\text{dist}(c_0, c_i) \leq \frac{\epsilon}{3z}(1+\epsilon)^{\frac{i}{z}}$.

*Proof.* For every $2 \leq i \leq t$, if $c_{i-1} = c_i$ then $\text{dist}(c_{i-1}, c_i) = 0$. If $c_{i-1} \neq c_i$, then $\text{avgcost}_z(\text{ball}_{\vec{P}}(c_i, (1+\epsilon)^{\frac{i}{z}}), c_i) \leq \frac{\epsilon^{2z}}{9^z z^z}(1+\epsilon)^i$. By letting $j = i-1$ in Fact B.2, we have

$$\frac{\epsilon^{2z}}{9^z z^z}(1+\epsilon)^i + \frac{\epsilon^{2z}}{9^z z^z}(1+\epsilon)^i \geq \text{avgcost}_z(\text{ball}_{\vec{P}}(c_i, (1+\epsilon)^{\frac{i}{z}}), c_{i-1}) + \text{avgcost}_z(\text{ball}_{\vec{P}}(c_i, (1+\epsilon)^{\frac{i}{z}}), c_i)$$

$$= \frac{1}{|\text{ball}_{\vec{P}}(c_i, (1+\epsilon)^i)|} \sum_{\vec{p} \in \text{ball}_{\vec{P}}(c_i, (1+\epsilon)^{\frac{i}{z}})} (\vec{P}(p)\, \text{dist}^z(p, c_{i-1}) + \vec{P}(p)\, \text{dist}^z(p, c_i))$$

$$\geq \frac{1}{|\text{ball}_{\vec{P}}(c_i, (1+\epsilon)^i)|} \sum_{\vec{p} \in \text{ball}_{\vec{P}}(c_i, (1+\epsilon)^{\frac{i}{z}})} (\frac{1}{2^{z-1}}\vec{P}(p)\, \text{dist}^z(c_{i-1}, c_i)) \geq \frac{\text{dist}^z(c_{i-1}, c_i)}{2^{z-1}}.$$

So we have $\text{dist}(c_{i-1}, c_i) \leq \frac{2\epsilon^2}{9z}(1+\epsilon)^{\frac{i}{z}}$. For the second part, it suffices to show that $\text{dist}(c_{i-1}, c_i) \leq (1+\epsilon)^{\frac{i}{z}} - (1+\epsilon)^{\frac{i-1}{z}}$. And this can be obtained from the first part. For the third part, we have $\text{dist}(c_0, c_i) \leq \sum_{j=1}^{i} \text{dist}(c_{j-1}, c_j) \leq \frac{2\epsilon^2}{9z}\frac{(1+\epsilon)^{\frac{1}{z}}(1+\epsilon)^{\frac{i}{z}}}{\epsilon} \leq \frac{\epsilon}{3z}(1+\epsilon)^{\frac{i}{z}}$. $\qquad\square$

**Lemma B.4.** *Let $(c_0, c_1, \ldots, c_t)$ be a robust sequence with respect to $\vec{P}$, and $\vec{P}_0 \subseteq \vec{P}$ be a subset of $\vec{P}$ such that $\text{ball}_{\vec{P}}(c_i, (1+\epsilon)^{\frac{i}{z}}) = \text{ball}_{\vec{P}_0}(c_i, (1+\epsilon)^{\frac{i}{z}})$ for all $i \in [t]$. Then $\forall i \in [t]$*

$$\text{cost}_z(\vec{P}_0, c_0) \leq (1+2\epsilon)\, \text{cost}_z(\vec{P}_0, c_i).$$

*Proof.* For every $1 \leq i \leq t-1$, let $\vec{D}_i := \text{ball}_{\vec{P}_0}(c_{i+1}, (1+\epsilon)^{\frac{i+1}{z}}) \setminus \text{ball}_{\vec{P}_0}(c_i, (1+\epsilon)^{\frac{i}{z}})$. Let $\vec{D}_t := \vec{P}_0 \setminus \text{ball}_{\vec{P}}(c_t, (1+\epsilon)^{\frac{t}{z}})$. By the third property of Lemma B.3, we have $\forall 1 \leq i \leq t, \forall \vec{p} \in \vec{D}_i$, $\text{dist}(c_i, c_0) \leq \frac{\epsilon}{3z}(1+\epsilon)^{\frac{i}{z}} \leq \frac{\epsilon}{3z}\text{dist}(c_i, p)$. So

$$\text{dist}(p, c_0)^z \leq (1+\epsilon)\, \text{dist}^z(p, c_i) + (\frac{2z+\epsilon}{\epsilon})^{z-1}\, \text{dist}^z(c_i, c_0) \leq (1+\epsilon + \frac{\epsilon(2z+\epsilon)^{z-1}}{(3z)^z})\, \text{dist}^z(p, c_i) \leq (1+2\epsilon)\, \text{dist}^z(p, c_i).$$

By Definition 4.1, in the point set $\text{ball}_{\vec{P}}(c_i, (1+\epsilon)^{\frac{i}{z}})$, we have

$$\text{cost}_z(\text{ball}_{\vec{P}_0}(c_i, (1+\epsilon)^{\frac{i}{z}}), c_i) \geq \text{cost}_z(\text{ball}_{\vec{P}_0}(c_i, (1+\epsilon)^{\frac{i}{z}}), c_{i-1}).$$

Now we combine the above inequalities to bound the cost of $p_0$ by the following inequalities:

$$\text{cost}_z(\vec{P}_0, c_i) = \text{cost}_z(\text{ball}_{\vec{P}_0}(c_i, (1+\epsilon)^{\frac{i}{z}}), c_i) + \text{cost}_z(\cup_{j \geq i}\vec{D}_j, c_i)$$

$$\geq \text{cost}_z(\text{ball}_{\vec{P}_0}(c_i, (1+\epsilon)^{\frac{i}{z}}), c_{i-1}) + \frac{1}{1+2\epsilon}\text{cost}_z(\cup_{j \geq i}\vec{D}_j, c_0) \qquad \text{(by Fact B.2)}$$

$$= \text{cost}_z(\text{ball}_{\vec{P}_0}(c_{i-1}, (1+\epsilon)^{\frac{i-1}{z}}), c_{i-1}) + \text{cost}_z(\vec{D}_{i-1}, c_{i-1}) + \frac{1}{1+2\epsilon}\text{cost}_z(\cup_{j \geq i}\vec{D}_j, c_0)$$

$$\geq \text{cost}_z(\text{ball}_{\vec{P}_0}(c_{i-1}, (1+\epsilon)^{\frac{i-1}{z}}), c_{i-1}) + \frac{1}{1+2\epsilon}(\text{cost}_z(\vec{D}_{i-1}, c_0) + \text{cost}_z(\cup_{j \geq i}\vec{D}_j, c_0))$$

$$\geq \ldots$$

$$\geq \text{cost}_z(\text{ball}_{\vec{P}_0}(c_1, (1+\epsilon)^{\frac{1}{z}}), c_0) + \frac{1}{1+2\epsilon}\text{cost}_z(\cup_{j \geq 1}\vec{D}_j, c_0)$$

$$\geq \frac{1}{1+2\epsilon}\text{cost}_z(\vec{P}_0, c_0).$$

$\qquad\square$

**Lemma B.5.** *Given a weighted point set $\vec{P}$ and a center set $C$ with witness mapping $\text{wit}_C$, during the whole process of $\text{MAKEROBUST}(\vec{P}, (C, \text{wit}_C))$, the size of center set $C$ remains the same as the input $C$.*

To prove the lemma, we first show the following technical lemma.

**Lemma B.6.** *Let $\vec{P}$ be a weighted point set, $(C, \mathrm{wit}_C)$ be a center set with witness mapping. Suppose there is $c \in C$ such that $\mathrm{wit}_C(c)$ is not $t_C(c)$-prefix robust with respect to $\vec{P}$, and let $c_0 := \mathrm{ROBUSTIFY}(\vec{P}, C, c)$. Then for every $\hat{c} \in C \setminus \{c\}$, $\mathrm{dist}(c_0, c) \leq \frac{\epsilon(1+\epsilon)^{\frac{1}{z}}}{15z} \mathrm{dist}(c, \hat{c})$. Furthermore $\mathrm{dist}(c_0, \hat{c}) \leq (1 + \frac{\epsilon(1+\epsilon)^{\frac{1}{z}}}{15z}) \mathrm{dist}(c, \hat{c})$.*

*Proof.* Let $c'$ be the closest point to $c$ in $C \setminus \{c\}$. By line 1 of ROBUSTIFY, $\mathrm{wit}_C(c_0)$ is $t$-prefix robust such that
$$5(1 + \epsilon)^{\frac{t-1}{z}} \leq \mathrm{dist}(c, C \setminus \{c\}) = \mathrm{dist}(c, c').$$

By Lemma B.3, we have $\mathrm{dist}(c_0, c) \leq \frac{\epsilon}{3z}(1 + \epsilon)^{\frac{t}{z}}$, combining the two inequalities we have for every $\hat{c} \in C \setminus \{c\}$, $\mathrm{dist}(c_0, c) \leq \frac{\epsilon(1+\epsilon)^{\frac{1}{z}}}{15z} \mathrm{dist}(c, c') \leq \frac{\epsilon(1+\epsilon)^{\frac{1}{z}}}{15z} \mathrm{dist}(c, \hat{c})$. By triangular inequality we have
$$\mathrm{dist}(c_0, \hat{c}) \leq \mathrm{dist}(c_0, c) + \mathrm{dist}(c, \hat{c}) \leq (1 + \frac{\epsilon(1+\epsilon)^{\frac{1}{z}}}{15z}) \mathrm{dist}(c, \hat{c}).$$
□

Now we are ready to prove Lemma B.5.

*Proof of Lemma B.5.* We prove this by induction. Let the size of the input center set be $n$. Our induction hypothesis is for every round of the while loop, the size of the center set $C$ in line 1 is $n$. The base case in the first round is naturally true by definition. For the inductive step, suppose at the $i$-th round for $i \geq 2$, at the beginning of the loop $|C| = n$. If ROBUSTIFY$(\vec{P}, C, c)$ is called for a center $c \in C$, let $(c_0, v) := \mathrm{ROBUSTIFY}(\vec{P}, C, c)$. Recall that by Lemma B.6, $\mathrm{dist}(c_0, c) \leq \frac{\epsilon(1+\epsilon)^{\frac{1}{z}}}{15z} \mathrm{dist}(c, C \setminus \{c\})$, so $c_0$ can not coincide with any center in $C \setminus \{c\}$. Then at the end of the loop, the size of the center set is still $n$. □

Next we argue that during the execution of MAKEROBUST, once a center is swapped out, it can never be swapped back again.

**Lemma B.7.** *Suppose MAKEROBUST is run on some point set $\vec{P}$ and center set with witness mapping $(C, \mathrm{wit}_C)$. During the execution of MAKEROBUST, if $(c_0, v) = \mathrm{ROBUSTIFY}(\vec{P}, C, c)$ for a center $c \in C$ and $c$ is replaced by $c_0$, then MAKEROBUST does not call ROBUSTIFY$(\vec{P}, C, c_0)$ until the end of the algorithm.*

*Proof.* To clearly track the changing during the algorithm, we construct $k$ queues $(Q_1, Q_2, \ldots, Q_k)$. The elements in the queues are points from the candidate center set with a witness. At first, each queue $Q_i$ contains one element in $C$ with the witness. During the execution, if $(c_0, v) = \mathrm{ROBUSTIFY}(c)$ is called, we find the unique queue $Q$ where $c$ is at the end of the queue. And add element $c_0$ to that queue with the witness $w$.

We first point out the above procedure is well-defined. This means that at any time, the end elements of these queues correspond to the center set. This can be concluded by Lemma B.5 because the size of the center set does not change and any center point must be at the end of one queue.

By our queue construction, for each round, the union of the end element for all queues is exactly the center set. Moreover, the elements in a queue totally capture how a center in the original center set is transferred. Leveraging these queues it suffices to show that none of the $k$ queues has a length more than 2.

Suppose for contradiction, there exists a queue $Q$ with a length of at least 3 during the execution of MAKEROBUST. We consider the first queue such that this case happens. Suppose the algorithm runs $s$ rounds and denote the immediate output set as $((C_1 := C, \mathrm{wit}_{C_1}), (C_2, \mathrm{wit}_{C_2}), \ldots, (C_s, \mathrm{wit}_{C_s}))$. Let $j$ be the minimum number such that at the $j + 1$-th round, there is a queue with a length 3.

Moreover, let $i$ be the minimum number such that at $i + 1$-th round, $c_0$ is in $Q$. Then by the property that each end element of a queue is in the center set for any round, $c_0$ keeps staying in the center set from $i + 1$-th round to $j$-th round.

Let $c'$ be the closest point in $C_i \setminus \{c\}$ to $c$. We note that $c'$ is also the closest point in $C_{i+1} \setminus \{c_0\}$.

Recall that from the definition, $v(c_0)$ is $t_{C_{i+1}}(c_0)$-prefix robust with respect to $P$, where $t_{C_{i+1}}(c_0)$ is the value defined in line 1 of Algorithn 2. We discuss the following two cases separately.

1. $c'$ is in the center set $C_{j+1}$. By Lemma B.6, we have

$$\operatorname{dist}(c_0, c') \leq \operatorname{dist}(c_0, c) + \operatorname{dist}(c, c') \leq (1 + \frac{\epsilon(1+\epsilon)^{\frac{1}{z}}}{15z}) \operatorname{dist}(c, c').$$

By the definition of $t_{C_{i+1}}(c_0)$ again we have

$$(1+\epsilon)^{t_{C_{i+1}}(c_0)} \geq \frac{\operatorname{dist}(c, c')}{5} \geq \frac{\operatorname{dist}(c_0, c')}{5(1 + \frac{\epsilon(1+\epsilon)^{\frac{1}{z}}}{15z})} \geq \frac{\operatorname{dist}(c_0, c')}{10} \geq \frac{\operatorname{dist}(c_0, C_{j+1} \setminus \{c_0\}}{10}.$$

Recall that $t_{C_{j+1}}(c_0)$ is the smallest integer $t$ such that $(1+\epsilon)^{\frac{t}{z}} \geq \operatorname{dist}(c_0, C_{j+1} \setminus \{c_0\})/10$. Then we have $t_{C_{i+1}}(c_0) \geq t_{C_{j+1}}(c_0)$ thus $w(c_0)$ is still $t_{C_{j+1}}(c_0)$-prefix robust. This contradicts our assumption that $v(c_0)$ is not $t_{C_{j+1}}(c_0)$-robust.

2. Otherwise, $c'$ is not in $C_{j+1}$. Let $Q'$ be the queue such that when calling ROBUSTIFY($\vec{P}, C_i, c$), $c'$ is at the end of $Q'$. If $c'$ is not in $C_{j+1}$, we know that there must be a center with witness $(c'_0, v'_0)$ being added to $Q'$ before $\hat{c}_0$ being in $Q$. On the other hand, as $c'$ is at the end of $Q'$ at the $i+1$-th round, so $c'_0$ has to be in $Q'$ after the $i+1$-th round. Suppose for a number $i < r < j$, $r+1$ is the minimum round such that $c'_0$ is in $Q'$. Recall that $c_0$ is at the end of $Q$ from the $i+1$-th round to the $r$-th round, so $c_0$ must be at the end of $Q$ at the $r$-th round thus $c_0$ is in the center set $C_r$.

   As at the $r$-th round, ROBUSTIFY($c'$) is called, so by Lemma B.6 $\operatorname{dist}(c', c'_0) \leq \frac{\epsilon(1+\epsilon)^{\frac{1}{z}}}{15z} \operatorname{dist}(c_0, c') \leq [(\frac{\epsilon(1+\epsilon)^{\frac{1}{z}}}{15z}) + (\frac{\epsilon(1+\epsilon)^{\frac{1}{z}}}{15z})^2] \operatorname{dist}(c, c')$. Thus we have

$$\operatorname{dist}(c_0, c'_0) \leq \operatorname{dist}(c_0, c') + \operatorname{dist}(c', c'_0) \leq (1 + \frac{\epsilon(1+\epsilon)^{\frac{1}{z}}}{15z})^2 \operatorname{dist}(c, c').$$

   Recall that by our selection, $Q$ is the first queue with 3 elements. So at the $j+1$-th round, $Q'$ has to have only 2 elements: $c'$ and $c'_0$. So $c'_0$ is in the center set $C_{j+1}$. As $c_0$ is $t(c_0)$-robust, we have

$$(1+\epsilon)^{t_{C_{i+1}}(c_0)} \geq \frac{\operatorname{dist}(c, c')}{5} \geq \frac{\operatorname{dist}(c_0, c'_0)}{5(1 + \frac{\epsilon(1+\epsilon)^{\frac{1}{z}}}{15z})^2} \geq \frac{\operatorname{dist}(c_0, c'_0)}{10} \geq \frac{\operatorname{dist}(c_0, C_{j+1} \setminus \{c_0\})}{10}.$$

   This also contradicts our assumption that $v(c_0)$ is not $t_{C_{j+1}}(c_0)$-prefix robust.

$\square$

**Lemma B.8.** *Let $\vec{P}$ and $\vec{P}'$ be two weighted point sets, $(c_0, c_1 \ldots, c_t)$ be a robust sequence with respect to points set $\vec{P}$. If $\operatorname{ball}_{\vec{P}}(c_0, (1 + \frac{\epsilon}{3z})(1 + \epsilon)^{\frac{t}{z}}) = \operatorname{ball}_{\vec{P}'}(c_0, (1 + \frac{\epsilon}{3z})(1 + \epsilon)^{\frac{t}{z}})$, then $(c_0, c_1 \ldots, c_t)$ is also robust with respect to point set $\vec{P}'$.*

*Proof.* By Lemma B.3, for every $0 \leq i \leq t - 1$, $\operatorname{ball}_{\vec{P}}(c_i, (1+\epsilon)^{\frac{i}{z}}) \subseteq \operatorname{ball}_{\vec{P}}(c_{i+1}, (1+\epsilon)^{\frac{i+1}{z}})$. So if $\operatorname{ball}_{\vec{P}}(c_t, (1+\epsilon)^{\frac{t}{z}}) = \operatorname{ball}_{\vec{P}'}(c_t, (1+\epsilon)^{\frac{t}{z}})$, then $(c_0, \ldots, c_t)$ is a $t$-robust sequence with respect to points set $\vec{P}'$. For every point $\vec{p} \in \operatorname{ball}_{\vec{P}}(c_t, (1+\epsilon)^t)$, by Lemma B.3 we have

$$\operatorname{dist}(c_0, p) \leq \operatorname{dist}(c_0, c_t) + \operatorname{dist}(c_t, p) \leq (1 + \frac{\epsilon}{3z})(1+\epsilon)^{\frac{t}{z}}.$$

So $\operatorname{ball}_{\vec{P}}(c_t, (1+\epsilon)^{\frac{t}{z}}) \subseteq \operatorname{ball}_{\vec{P}}(c_0, (1 + \frac{\epsilon}{3z})(1+\epsilon)^{\frac{t}{z}})$. And this suffices to prove the lemma. $\square$

## C ANALYSIS OF MAIN ALGORITHM IN SECTION 4.2

Below we first define some notations used in the proof for the convenience of presentation. For each phase $e$, suppose the length of the phase is $\ell_e + 1$, we write $\vec{P}_0^{(e)}$, and $\vec{P}_{end}^{(e)}$ as the points set at the beginning and the end of the phase, and $(U_0^{(e)}, U_1^{(e)}, \ldots, U_{\ell_e}^{(e)}, U_{end}^{(e)} := U_{\ell_e+1}^{(e)})$ as the intermediate center set during phase $e$. The intermediate center after swapping is denoted as $W^{(e)}$.

## C.1 TIME COMPLEXITY

The running time of the algorithm is summarized in the following lemma.

**Lemma C.1.** *The consistent* $(k, z)$*-CLUSTERING algorithm for bounded input runs in* $O(k^2 mT(m) + dkm^2 2^{\mathrm{poly}(\epsilon^{-1})} \log(\Delta))$ *time.*

*Proof.* We first analyze the time complexity for a fixed phase. Recall that each phase consists of four steps and we give the time complexity for each step.

**Deleting centers.** In this step, the algorithm enumerate all possible number $0 \le \ell < k$, and then call the $\alpha$-approximate algorithm on the candidate center set $U_0$ with center set size less than $k - \ell$. The time complexity is $kT(m)$.

**Handling insertions.** The time complexity is $O(\ell k d)$, wince we need to check if the current center belongs to the current center set which has at most $k$ points.

**Swapping centers.** In this step, following the proof of Lemma C.2, it suffices to consider the time to compute an $\alpha$-approximate solution and then find all valid well separated pairs. The time complexity is $O(k^2 + T(m))$ where the term $k^2$ comes from computing the pair-wise distance between two center sets.

**Robustifying centers.** In this step, the algorithm runs subroutine MAKEROBUST(Algorithm 3), which runs subroutine ROBUSTIFY(Algorithm 2) if a center is not bounded robust. Since checking if a center with witness is bounded robust takes the same order of time as computing a robust sequence, it suffices to analyze the running time of ROBUSTIFY.

When computing the robust sequence in line 2 of ROBUSTIFY, the algorithm needs to compute a near-optimal center for $\mathrm{ball}_{\vec{P}}(c_i, (1 + \epsilon)^{\frac{i}{z}})$ for some $i$ in Definition 4.1, which can be done in $O(dm2^{\mathrm{poly}(\epsilon^{-1})})$ time by Chen (2009). As the length of a robust sequence can be at most $O(\log(\Delta)/\epsilon)$ and ROBUSTIFY is called at most $k$ times by Lemma B.7, the running time for $k$ calls of ROBUSTIFY is $O(km2^{\mathrm{poly}(\epsilon^{-1})} \log(\Delta))$.

Finally, since the number of phases is upper bounded by the total number of inputs which we denoted as $m$, the time complexity is $O(k^2 mT(m) + dkm^2 2^{\mathrm{poly}(\epsilon^{-1})} \log(\Delta))$ in total for all phases combined. □

## C.2 COST ANALYSIS

We analyze the competitive ratio in this section. Recall that the algorithm runs in phases, and in each phase $e$, the algorithm starts from some center set $U_0^{(e)}$. Hence, our plan is to inductively show that $U_0^{(e)}$ is $(1 + O(\epsilon))\alpha$-approximate. Notice that this alone is not enough, since for all but the last input point during a phase, the output center set is derived from $U_0^{(e)}$ (by first deleting some centers from it and then adding some input points), so we also need to analyze the ratio for these intermediate steps, and we show it is $(1 + O(\epsilon))\alpha^2$-approximate. We also notice that $U_0^{(e)}$ is changed only in the last step of a phase which particularly has a swapping procedure (and hence the induction is to analyze this last step only).

Now, observe that to show $U_0^{(e)}$ is $(1 + O(\epsilon))\alpha$-approximate, it suffices to bound the approximation ratio after swapping centers, since if we can show the center set $W^{(e)}$ resulted from the swap is $(1 + O(\epsilon))\alpha$-approximate, then by Lemma B.4, the center set $U_{end}^{(e)}$ is still a $(1 + O(\epsilon))\alpha$-approximate solution. To this end, recall that in the algorithm in Section 4.2, $W^{(e)}$ is obtained by swapping out centers in $U_0^{(e)}$ that either do not form a $\frac{\epsilon^4}{200z}$-well separated pair with a $\alpha$-approximate solution for $\vec{P}_{\ell+1}$, or serve points in $\vec{P}_{end}^{(e)} \setminus \vec{P}_0^{(e)}$. In the following technical lemma, we show that this swapping rule indeed guarantees a $(1 + 5\epsilon)\alpha$-approximate ratio.

**Lemma C.2.** *Let $U$ and $V$ be two center sets, and $\vec{P}$, $\vec{P}'$ be two weighted point sets such that $\vec{P} \subseteq \vec{P}'$. Suppose $U$ is equipped with witness mapping $\mathrm{wit}_U$ and $(U, \mathrm{wit}_U)$ is bounded robust with respect to $\vec{P}$. Given $\epsilon \in (0, 1)$, let $s \geq 1$ be any integer such that the following hold:*

1. $(u_1, v_1), (u_2, v_2) \ldots (u_s, v_s)$ *are* $\frac{\epsilon^4}{400z}$*-well separated pairs.*
2. $\vec{P}'[V, v_i] \subseteq \vec{P}$ *for* $i \in [s]$.

*Let $C := \{u_1, u_2, \ldots, u_s, v_{s+1}, \ldots, v_k\}$, then*

$$\mathrm{cost}_z(\vec{P}', C) \leq (1 + 5\epsilon)\,\mathrm{cost}_z(\vec{P}', V).$$

Assume this lemma is true (and the proof appears after this part of argument), we can conclude the competitive ratio analysis as follows.

**Lemma C.3.** *The consistent algorithm for bounded input is $(1 + 233\epsilon)\alpha^2$-competitive.*

*Proof.* We first prove for every phase $e$, $(U_0^{(e)}, \mathrm{wit}_{U_0^{(e)}})$ is bounded robust with respect to $\vec{P}_0^{(e)}$ and $U_0^{(e)}$ is a $(1 + 17\epsilon)\alpha$-approximate solution for $\vec{P}_0^{(e)}$. This is done by induction on the phase, the base case is the first phase, where the algorithm runs the $\alpha$-approxiamte algorithm and then calls MAKEROBUST for the output to obtain $U_0$. By Lemma B.4, $U_0$ is a $(1 + 2\epsilon)\alpha$-approximate solution. The bounded robust property naturally holds since $U_0$ is the output of MAKEROBUST. For the inductive step, suppose for phase $e$, the induction hypothesis is true. Let $V$ be an $\alpha$-approximate solution for $\vec{P}_{end}^{(e)}$. By letting $\vec{P}_0^{(e)}$ and $\vec{P}_{end}^{(e)}$ be the corresponding $\vec{P}$ and $\vec{P}'$, $U_0^{(e)}$, $V$ and $W^{(e)}$ be the corresponding $U$, $V$ and $C$ in Lemma C.2, we know $W^{(e)}$ is a $(1 + 5\epsilon)\alpha$-approximate solution for $P_{end}^{(e)}$. By Lemma B.4, calling MAKEROBUST$(\vec{P}_{end}^{(e)}, (W^{(e)}, \mathrm{wit}_{W^{(e)}}))$ increases the cost by a factor of $1 + 2\epsilon$, so $U_{end}^{(e)}$ is a $(1 + 17\epsilon)\alpha$-approximate solution. Moreover, $U_{end}^{(e)}$ is naturally bounded robust because it is the output of MAKEROBUST. Thus we finish the induction.

Now we prove the competitive ratio for the algorithm. For every phase $e$, we know $U_0^{(e)}$ is a $(1 + 17\epsilon)\alpha$-approximate solution. Suppose phase $e$ consists $\ell_e$ insertions for some $\ell \geq 1$. For all $1 \leq i \leq \ell_e$, we have

$$\mathrm{cost}_z(\vec{P}_i^{(e)}, U_i^{(e)}) \leq (1 + 12\epsilon)\alpha\,\mathrm{cost}_z(\vec{P}_0^{(e)}, U_0^{(e)})$$
$$\leq (1 + 17\epsilon)(1 + 12\epsilon)\alpha^2\,\mathrm{OPT}(\vec{P}_0^{(e)})$$
$$\leq (1 + 17\epsilon)(1 + 12\epsilon)\alpha^2\,\mathrm{OPT}(\vec{P}_i^{(e)}) \leq (1 + 233\epsilon)\alpha^2\,\mathrm{OPT}(\vec{P}_i^{(e)}),$$

where the first inequality comes from the fact that $\mathrm{cost}_z(\vec{P}_i^{(e)} \setminus \vec{P}_0^{(e)}, U_i^{(e)}) = 0$ and the definition of the algorithm. The second inequality follows from $U_0^{(e)}$ is a $(1 + 17\epsilon)\alpha$-approximate solution for $\vec{P}_0^{(e)}$. $\qquad\square$

Now we give the proof of our key lemma Lemma C.2.

*Proof of Lemma C.2.* As the second condition $\vec{P}'[V, v_i] \subseteq \vec{P}$ is equivalent to $\vec{P}'[V, v_i] = \vec{P}[V, v_i]$, we write

$$\mathrm{cost}_z(\vec{P}', C) = \sum_{i \leq s} \mathrm{cost}_z(\vec{P}'[C, u_i], u_i) + \sum_{i \geq s+1} \mathrm{cost}_z(\vec{P}'[C, v_i], v_i)$$
$$\leq \sum_{i \leq s} \mathrm{cost}_z(\vec{P}'[V, v_i], u_i) + \sum_{i \geq s+1} \mathrm{cost}_z(\vec{P}'[V, v_i], v_i)$$

The inequality holds because it's optimal to assign every point in $\vec{P}'$ to the closest center in $C$.

We will show that for a well separated pair $(u_i, v_i)$, $i \leq s$, $\mathrm{cost}_z(\vec{P}'[V, v_i], u_i) \leq (1 + 2\epsilon)\,\mathrm{cost}_z(\vec{P}'[V, v_i], v_i)$ and this suffices.

Suppose $(u, v)$ is a well separated pair, wit$_U(u) = (u, c_1, c_2, \ldots, c_{t(u)})$, then we have

$$\text{dist}(u, v) \leq \frac{\epsilon^4}{400z} \text{dist}(u, U \setminus \{u\}) \leq \frac{\epsilon^4}{40z}(1 + \epsilon)^{\frac{t(u)}{z}}$$

which follows by the Definition 4.3 and Definition 4.2.

So there exists a $t^* \leq t(u)$ such that $\frac{\epsilon^4}{40z}(1 + \epsilon)^{\frac{t^*-1}{z}} \leq \text{dist}(u, v) \leq \frac{\epsilon^4}{40z}(1 + \epsilon)^{\frac{t^*}{z}}$.

**Claim C.4.** $\text{ball}_{\vec{P}}(u, (1 + \frac{\epsilon}{3z})(1 + \epsilon)^{\frac{t^*}{z}}) \subseteq \vec{P}[V, v]$.

*Proof.* Let $q \in \text{ball}_{\vec{P}}(u, (1 + 3\epsilon)(1 + \epsilon)^{\frac{t^*}{z}})$, then we have

$$\text{dist}(q, v) \leq \text{dist}(q, p_{t^*}) + \text{dist}(c_{t^*}, u) + \text{dist}(u, v) \leq (1 + \frac{\epsilon}{3z})(1 + \epsilon)^{\frac{t^*}{z}} + \frac{\epsilon}{3z}(1 + \epsilon)^{\frac{t^*}{z}} + \frac{\epsilon^4}{40z}(1 + \epsilon)^{\frac{t^*}{z}} \leq (1 + \epsilon)(1 + \epsilon)^{\frac{t^*}{z}}$$

where the second term of the second inequality follows by Lemma B.3. On the other hand, for any $v' \subseteq V$ such that $v' \neq v$, we have

$$\text{dist}(q, v') \geq \text{dist}(v', v) - \text{dist}(v, q) \geq \frac{400z \, \text{dist}(u, v)}{\epsilon^4} - \text{dist}(q, v) \geq \text{dist}(q, v),$$

where the final inequality follows by $\frac{\epsilon^4}{40z}(1 + \epsilon)^{\frac{t^*-1}{z}} \leq \text{dist}(u, v)$ and $\text{dist}(q, v) \leq (1 + \epsilon)(1 + \epsilon)^{\frac{t^*}{z}}$. So $v$ is the nearest center to $q$ in $V$. This finishes the proof of Claim C.4. $\square$

As $(u, v)$ is close enough, for the points outside the ball $\text{ball}_{\vec{P}}(u, (1 + \frac{\epsilon}{3z})(1 + \epsilon)^{\frac{t^*}{z}})$, the cost induced by $u$ is naturally a good approximation for the optimal cost. As long as the average cost in the small ball $\text{ball}_{\vec{P}}(u, (1 + \frac{\epsilon}{3z})(1 + \epsilon)^{\frac{t^*}{z}})$ around $v$ is large enough, then $\text{cost}_z(\text{ball}_{\vec{P}}(u, (1 + \frac{\epsilon}{3z})(1 + \epsilon)^{\frac{t^*}{z}}), u)$ is a good approximation for $\text{cost}_z(\text{ball}_{\vec{P}}(u, (1 + \frac{\epsilon}{3z})(1 + \epsilon)^{\frac{t^*}{z}}), v)$. Summing the two parts together achieves our goal.

To formalize the above discussion, we handle two situations $c_{t^*} = c_{t^*-1}$ and $c_{t^*} \neq c_{t^*-1}$ separately.

If $c_{t^*} = c_{t^*-1}$, this means $\text{avgcost}(\text{ball}_{\vec{P}_0}(c_{t^*}, (1 + \epsilon)^{\frac{t^*}{z}}), c_{t^*}) \geq 2^{z-1} \frac{\epsilon^{2z}}{(9z)^z}(1 + \epsilon)^{\frac{t^*}{z}}$. We first show that the average cost of $u$ in $\text{ball}_{\vec{P}_0}(u, (1 + \epsilon)^{\frac{t^*+1}{z}})$ can not be too small.

**Claim C.5.** $\text{avgcost}_z(\text{ball}_{\vec{P}}(u, (1 + \frac{\epsilon}{3z})(1 + \epsilon)^{\frac{t^*}{z}}), u) \geq \frac{\epsilon^{2z+1}}{2 \cdot (18z)^z}(1 + \epsilon)^{t^*}$.

*Proof.* If $u = c_{t^*}$ then we naturally have

$$\text{avgcost}_z(\text{ball}_{\vec{P}}(u, (1 + \frac{\epsilon}{3z})(1 + \epsilon)^{\frac{t^*}{z}}), u) = \text{avgcost}_z(\text{ball}_{\vec{P}}(c_{t^*}, (1 + \epsilon)^{\frac{t^*}{z}}), c_{t^*}) \geq \frac{\epsilon^{2z}}{(9z)^z}(1 + \epsilon)^{t^*}.$$

So from now on we assume $u \neq c_{t^*}$. Suppose $c_{t^*} = c_{t^*-1} = \ldots = c_{t'} \neq c_{t'-1}$ for some $t' > 0$. Then by definition we have

$$\text{avgcost}_z(\text{ball}_{\vec{P}}(c_{t^*}, (1 + \epsilon)^{\frac{t'}{z}}), c_{t^*}) \leq \frac{\epsilon^{2z}}{(9z)^z}(1 + \epsilon)^{t'}.$$

Let $\vec{S}_1$ be the points set $\text{ball}_{\vec{P}}(c_{t^*}, (1 + \epsilon)^{\frac{t^*}{z}}) \setminus \text{ball}_{\vec{P}}(c_{t^*}, (1 + \epsilon)^{\frac{t'}{z}})$, $\vec{S}_2$ be the points set $\text{ball}_{\vec{P}}(c_{t^*}, (1 + \epsilon)^{\frac{t'}{z}})$ and $\vec{Q}$ be the points set $\text{ball}_{\vec{P}}(u, (1 + \frac{\epsilon}{3z})(1 + \epsilon)^{\frac{t^*}{z}})) \setminus (S_1 \cup S_2)$. Then we have

$$\frac{w(\vec{S}_1) \text{avgcost}_z(\vec{S}_1, p_{t^*}) + w(\vec{S}_2) \text{avgcost}_z(\vec{S}_2, p_{t^*})}{w(\vec{S}_1) + w(\vec{S}_2)} = \text{avgcost}_z(\text{ball}_{\vec{P}}(c_{t^*}, (1 + \epsilon)^{t^*}), c_{t^*}) \geq \frac{\epsilon^{2z}}{(9z)^z}(1 + \epsilon)^{t^*}.$$

Thus

$$\frac{w(\vec{S_1})\operatorname{cost}_z(\vec{S_1}, c_{t^*})}{w(\vec{S_1}) + w(\vec{S_2})} \geq \frac{\epsilon^{2z}}{(9z)^z}(1+\epsilon)^{t^*} - \operatorname{avgcost}_z(\vec{S_2}, c_{t^*}) \geq \frac{\epsilon^{2z}}{(9z)^z}((1+\epsilon)^{t^*} - (1+\epsilon)^{t'}) \geq \frac{\epsilon^{2z+1}(1+\epsilon)^{t^*}}{(9z)^z(1+\epsilon)}.$$

On the other hand, for every point $\vec{p}$ in $\vec{S_1}$, we have

$$\operatorname{dist}^z(p, c_{t^*}) = \operatorname{dist}^z(p, c_{t'}) \leq 2^{z-1}\operatorname{dist}^z(p, u) + 2^{z-1}\operatorname{dist}^z(c_{t'}, u) \leq 2^{z-1}(1 + \frac{\epsilon^z}{(3z)^z})\operatorname{dist}^z(p, u) \leq 2^z\operatorname{dist}^z(p, u).$$

$$(2)$$

Finally we have

$$\operatorname{avgcost}(\operatorname{ball}_{\vec{P}}(u, 1 + \frac{\epsilon}{3z})^{\frac{1}{z}}(1+\epsilon)^{\frac{t^*}{z}}, u)$$

$$= \frac{w(\vec{S_1})\operatorname{avgcost}(\vec{S_1}, u) + w(\vec{S_2})\operatorname{avgcost}(\vec{S_2}, u) + w(\vec{Q})\operatorname{avgcost}(\vec{Q}, u)}{w(\vec{S_1}) + w(\vec{S_2}) + w(\vec{Q})}$$

$$\geq \frac{w(\vec{S_1})\operatorname{avgcost}(\vec{S_1}, u) + w(\vec{Q})\operatorname{avgcost}(\vec{Q}, u)}{w(\vec{S_1}) + w(\vec{S_2}) + w(\vec{Q})}$$

$$\geq \frac{w(\vec{S_1})\operatorname{avgcost}(\vec{S_1}, u)}{w(\vec{S_1}) + w(\vec{S_2})}$$

$$\geq \frac{w(\vec{S_1})\operatorname{avgcost}(\vec{S_1}, c_{t^*})}{2^z(w(\vec{S_1}) + w(\vec{S_2}))}$$

$$\geq \frac{\epsilon^{2z+1}}{(18z)^z(1+\epsilon)}(1+\epsilon)^{t^*}$$

$$\geq \frac{\epsilon^{2z+1}}{2 \cdot (18z)^z}(1+\epsilon)^{t^*},$$

where the second inequality follows by $\operatorname{avgcost}(Q, u) \geq \operatorname{avgcost}(S_1, u)$, the third inequality follows by Equation (2). This finished the proof of Claim C.5. $\square$

Now we divide $\operatorname{cost}_z(\vec{P}[V, v], v)$ into two parts: the small ball $\operatorname{ball}_{\vec{P}}(u, (1 + \frac{\epsilon}{3z})(1+\epsilon)^{\frac{t^*}{z}})$ and the points outside the ball. Namely,

$$\operatorname{cost}_z(\vec{P}[V, v], v) = \operatorname{cost}_z(\operatorname{ball}_{\vec{P}}(u, (1 + \frac{\epsilon}{3z})(1+\epsilon)^{\frac{t^*}{z}}), v) + \operatorname{cost}_z(\vec{P}[V, v]\backslash\operatorname{ball}_{\vec{P}}(u, (1 + \frac{\epsilon}{3z})(1+\epsilon)^{\frac{t^*}{z}}), v)$$

We bound the two terms separately, for the first part.

$$\operatorname{cost}_z(\operatorname{ball}_{\vec{P}}(u, (1 + \frac{\epsilon}{3z})(1+\epsilon)^{\frac{t^*}{z}}), v)$$

$$= \sum_{\vec{p}\in\operatorname{ball}_{\vec{P}}(u,(1+\frac{\epsilon}{3z})(1+\epsilon)^{\frac{t^*}{z}})} [\vec{P}(p)\operatorname{dist}^z(p, v)] \geq \sum_{\vec{p}\in\operatorname{ball}_{\vec{P}}(u,(1+\frac{\epsilon}{3z})(1+\epsilon)^{\frac{t^*}{z}})} [\frac{\vec{P}(p)\operatorname{dist}^z(p, u)}{2^{z-1}} - \vec{P}(p)\operatorname{dist}^z(u, v)]$$

$$= |\operatorname{ball}_{\vec{P_0}}(u, (1 + \frac{\epsilon}{3z})(1+\epsilon)^{\frac{t^*}{z}})|(\frac{\operatorname{avgcost}(\operatorname{ball}_{\vec{P}}(u, (1 + \frac{\epsilon}{3z})(1+\epsilon)^{\frac{t^*}{z}}), u)}{2^{z-1}} - \operatorname{dist}^z(u, v))$$

$$\geq |\operatorname{ball}_{\vec{P_0}}(u, (1 + \frac{\epsilon}{3z})(1+\epsilon)^{\frac{t^*}{z}})|((1 - \epsilon)\operatorname{avgcost}(\operatorname{ball}_{\vec{P}}(u, (1 + \frac{\epsilon}{3z})(1+\epsilon)^{\frac{t^*}{z}}), u))$$

$$= (1 - \epsilon)\operatorname{cost}_z(\operatorname{ball}_{\vec{P}}(u, (1 + \frac{\epsilon}{3z})(1+\epsilon)^{\frac{t^*}{z}}), u).$$

where the second inequality follows by Claim C.5 and $\operatorname{dist}(u, v) \leq \frac{\epsilon^4}{40z}(1+\epsilon)^{\frac{t^*}{z}}$.

Next we deal with the other part. For any $p \in \vec{P}[V, v] \backslash \operatorname{ball}_{\vec{P}}(u, (1 + \frac{\epsilon}{3z})^{\frac{1}{z}}(1+\epsilon)^{\frac{t^*}{z}})$, we have

$$\operatorname{dist}^z(p, v) \geq \frac{\operatorname{dist}^z(p, u)}{2^{z-1}} - \operatorname{dist}^z(u, v) \geq (1 - \epsilon)\operatorname{dist}^z(p, u),$$

thus

$$\text{cost}_z(\vec{P}[V,v]\setminus \text{ball}_{\vec{P}}(u,(1+\frac{\epsilon}{3z})^{\frac{1}{z}}(1+\epsilon)^{\frac{t^*}{z}}),v) \geq (1-\epsilon)\,\text{cost}_z(\vec{P}[V,v]\setminus(1+\frac{\epsilon}{3z})^{\frac{1}{z}}(1+\epsilon)^{\frac{t^*}{z}}),u).$$

Now we add the two parts:

$$\text{cost}_z(\vec{P}[V,v],v) \geq (1-\epsilon)\,\text{cost}_z(\vec{P}[V,v],u)$$

So in this case $c_{t^*} = c_{t^*-1}$, we have

$$\text{cost}_z(\vec{P}[V,v],u) \leq \frac{1}{1-\epsilon}\,\text{cost}_z(\vec{P}[V,v],v) \leq (1+2\epsilon)\,\text{cost}_z(\vec{P}[V,v],v).$$

If $c_{t^*} \neq c_{t^*-1}$, this means $c_{t^*-1}$ is a near-optimal center for $\text{ball}_{\vec{P}}(c_{t^*},(1+\epsilon)^{\frac{t^*}{z}})$, thus we have

$$\text{cost}_z(\vec{P}[V,v],u) = \text{cost}_z(\text{ball}_{\vec{P}}(c_{t^*-1},(1+\epsilon)^{\frac{t^*}{z}}),u) + \text{cost}_z(\vec{P}[V,v]\setminus\text{ball}_{\vec{P}_0}(c_{t^*-1},(1+\epsilon)^{\frac{t^*}{z}}),u)$$

$$\leq (1+2\epsilon)\,\text{cost}_z(\text{ball}_{\vec{P}}(c_{t^*-1},(1+\epsilon)^{\frac{t^*}{z}}),c_{t^*-1}) + (1+\epsilon)\,\text{cost}_z(\vec{P}[V,v]\setminus\text{ball}_{\vec{P}_0}(c_{t^*-1},(1+\epsilon)^{\frac{t^*}{z}}),v)$$

$$\leq (1+2\epsilon)(1+\epsilon)\,\text{cost}_z(\text{ball}_{\vec{P}}(c_{t^*-1},(1+\epsilon)^{\frac{t^*}{z}}),v) + (1+\epsilon)\,\text{cost}_z(\vec{P}[V,v]\setminus\text{ball}_{\vec{P}_0}(c_{t^*-1},(1+\epsilon)^{\frac{t^*}{z}}),v)$$

$$\leq (1+5\epsilon)\,\text{cost}_z(\vec{P}[V,v],v)$$

where the first term of the first inequality follows by Lemma B.4, and the second term follows by $\text{dist}^z(p,v) \geq (1-\epsilon)\,\text{dist}^z(p,u)$ for $\vec{p} \in \vec{P}[V,v]\setminus\text{ball}_{\vec{P}}(c_{t^*-1},(1+\epsilon)^{\frac{t^*}{z}})$. The second inequality follows by $c_{t^*-1}$ is a $(1+\epsilon)$-approximation of the optimal $(1,z)$-clustering problem for the ball. This finishes the proof of Lemma C.2. $\qquad\square$

## C.3 Consistency Analysis

In this section we prove the consistency of our algorithm, summarized as the following lemma.

**Lemma C.6.** *Given an input stream of $m$ weighted points, the total consistency of the algorithm is* $O(\frac{\alpha 2^{O(z\log(z))}m\log^2(\Delta)}{\epsilon^{8z-2}})$.

By the definition of the algorithm, the consistency is upper bounded by $\sum_e \sum_{1\leq i\leq \ell_e}(|U_i^{(e)}\setminus U_{i-1}^{(e)}|) + |U_{end}^{(e)}\setminus U_{\ell_e}^{(e)}|$, which accounts for the changes of the center set both within phases and across phases. We first deal with the easy part: the consistency contributed by $\sum_{1\leq i\leq \ell_e}|U_i^{(e)}\setminus U_{i-1}^{(e)}|$, which is the consistency within phases. In this part, it is straightforward that the consistency is $\ell_e$. Note that a phase contains $\ell_e + 1$ insertions, so the amortized consistency during a phase is $O(1)$. Summing up all phases, the total consistency contributed by this part is $O(m)$.

Next we bound the consistency contributed by $|U_{end}^{(e)}\setminus U_{\ell_e}^{(e)}|$, which is the consistency across phases. First we have $|U_{end}^{(e)}\setminus U_{\ell_e}^{(e)}| \leq |U_{end}^{(e)}\setminus U_0^{(e)}| + \ell_e$. So it suffices to consider the term $|U_{end}^{(e)}\setminus U_0^{(e)}|$. Based on this observation, We partition $U_{end}^{(e)}$ into two parts.

1. **Updated centers**. We define the updated center set as $\text{UP}^{(e)}$ which consist of (1) centers in $U_{end}^{(e)}\cap(W^{(e)}\cap U_0^{(e)})$ and (2) obtained by calling $\textsc{Robustify}(\vec{P}_{end}^{(e)},W^{(e)},c)$ where $c \in W^{(e)}\cap U_0^{(e)}$. Centers in $\text{UP}^{(e)}$ are called updated centers.
2. **Fresh centers**. We define the fresh center set as $F^{(e)}$, which consists of the remaining centers, specifically, (1) the centers in $U_{end}^{(e)}\cap(W^{(e)}\setminus U_0^{(e)})$ and (2) centers that are obtained by calling $\textsc{Robustify}$ to centers in $W^{(e)}\setminus U_0^{(e)}$. Centers in $F^{(e)}$ are called fresh centers.

Based on the partition strategy, $U_{end}^{(e)}\setminus U_0^{(e)} \subseteq F^{(e)} \cup (\text{UP}^{(e)}\setminus U_0)$ so $|U_{end}^{(e)}\setminus U_0^{(e)}| \leq |F^{(e)}| + |\text{UP}^{(e)}\setminus U_0|$. So we can split the consistency across the whole phases $\sum_e |U_{end}^{(e)}\setminus U_0^{(e)}|$ into two parts:

$$\sum_e |U_{end}^{(e)}\setminus U_0^{(e)}| \leq \sum_e \left[|F^{(e)}| + |\text{UP}^{(e)}\setminus U_0^{(e)}|\right]. \qquad (3)$$

So it suffices to bound the two terms on the right hand side. For the fresh centers, the following lemma provides the bound.

**Lemma C.7.** $\sum_e |F^{(e)}| \leq O(\frac{\alpha 2^{O(z \log(z))} m}{\epsilon^{8z-3}})$.

*Proof.* Fix a phase $e$, note that the number of fresh centers in $e$ is upper bounded by the number of centers in $W^{(e)} \setminus U_0^{(e)}$. This is because for every $c \in W^{(e)} \setminus U_0^{(e)}$, either $c \in U_{end}^{(e)}$ or ROBUSTIFY$(\vec{P}_{end}^{(e)}, W^{(e)}, c) \in U_{end}^{(e)}$.

Let $V$ be an $\alpha$-approximate solution for $\vec{P}_{end}^{(e)}$. $s$ be the number such that for every $i \in [s]$, $(u_i, v_i)$ forms a $\frac{\epsilon^4}{400z}$-well separated pair and $\vec{P}_{end}^{(e)}[v_i] \subseteq \vec{P}_0$. Recall that the consistency algorithm for bounded size in Section 4.2 swap $v_{s+1}, \ldots, v_k$ into $U_0$ to produce $W$. So $|W^{(e)} \setminus U_0^{(e)}| = k - s$. On the other hand, suppose the number of $\frac{\epsilon^4}{400z}$-well-separated pairs between $U_0$ and $V$ is $t$, then $s$ is larger than $t - \ell_e - 1$ because at most $\ell_e + 1$ centers serve the $\ell_e + 1$ new points. Thus $|W^{(e)} \setminus U_0^{(e)}| \leq k - t + \ell_e + 1$.

The following technical lemma gives the lower bound for $t$, whose proof is provided in the next subsetion.

**Lemma C.8.** $U_0$ and $V$ have at least $k - O(\alpha 2^{O(z \log(z))} \ell / \epsilon^{8z-3})$ $\frac{\epsilon^4}{400z}$-well separated pairs.

*Proof.* The proof can be found in Section C.3.1. $\qquad\square$

By Lemma C.8, $|W^{(e)} \setminus U_0^{(e)}| \leq O(\alpha 2^{O(z \log(z))} \ell / \epsilon^{8z-3})$. So we also have $|F^{(e)}| \leq O(\frac{\alpha 2^{O(z \log(z))} \ell_e}{\epsilon^{8z-3}})$. Now summing over all phases, we get the number of new centers is at most the total number of $O(\frac{\alpha 2^{O(z \log(z))} m}{\epsilon^{8z-3}})$. This finishes the proof of Lemma C.7. $\qquad\square$

Next we consider the number of updated centers. Recall that we are in fact considering points in the difference of sets: $UP^{(e)} \setminus U_0^{(e)}$ for each phase $e$. We should point out that for a center $u$, there can be several phases $e$ such that $u \in UP^{(e)} \setminus U_0^{(e)}$. In this case, it means that $u$ is moved out from the center set and swapped back several times, and every time $u$ is counted independently.

Now we fix a phase $e$. Let $u$ be a center point in $UP^{(e)}$ (which is also in $U_0^{(e)}$) with witness $v(u)$ such that ROBUSTIFY of $u$ is called during the execution of MAKEROBUST$(\vec{P}_{end}, (W, \text{wit}_W))$. Suppose $v(u) = (u, p_1, p_2, \ldots, p_{t(u)})$ is the witness for some $t(u) \in \mathbb{N}$.

Note that duing the process of MAKEROBUST, the center keeps changing. Let $\hat{C}$ be the center set immediately before calling ROBUSTIFY for $u$. Then $v(u)$ is not $t_{\hat{C}}(u)$-prefix robust with respect to $\vec{P}_{end}^{(e)}$. By Definition 4.1, there are two cases:

1. $t(u) < t_{\hat{C}}(u)$ regardless of whether $v(u)$ is robust or not. In this case, we say $u$ is violated by centers update.

2. $t(u) \geq t_{\hat{C}}(u)$, but $v(u)$ is not $t_{\hat{C}}$-prefix robust with respect to $\vec{P}_{end}^{(e)}$. In this case, we say $u$ is violated by points insertion.

For the second case, as $u \in U_0^{(e)}$ and $U_0^{(e)}$ is bounded robust, we know that $v(u)$ is $t(u)$-robust with respect to points set $\vec{P}_0^{(e)}$. By the contradictory proposition of Lemma B.8, there must be a point $\vec{p} \in \vec{P}_{end}^{(e)} \setminus \vec{P}_0^{(e)}$, such that

$$\text{dist}(p, u) \leq (1 + \frac{\epsilon}{3z})(1 + \epsilon)^{\frac{t_{\hat{C}}(u)}{z}} \leq (1 + \frac{\epsilon}{3z})(1 + \epsilon)^{\frac{t(u)}{z}}. \tag{4}$$

Given a center point $u \in UP^{(e)}$ and a point $\vec{p} \in \vec{P}_{end}^{(e)} \setminus \vec{P}_0^{(e)}$, if $\text{dist}(u, p) \leq (1 + \frac{\epsilon}{3z})(1 + \epsilon)^{\frac{t(u)}{z}}$, then we say $\vec{p}$ violates $u$. Note that if a center $u$ is violated by points insertion then there must be at least one point that violates $u$. However, if a point violates $u$, $u$ may not be violated by points insertion.

We have the following bound for the two cases.

**Lemma C.9.** *For every $e$, every point $p \in \vec{P}_{end}^{(e)} \setminus \vec{P}_0^{(e)}$ violates at most $O(\log(\Delta))$ centers in $\mathrm{UP}^{(e)}$.*

*Proof.* Suppose $\vec{p} \in \vec{P}_{end}^{(e)} \setminus \vec{P}_0^{(e)}$ for a phase $e$. Let $s$ be the number of centers in $\mathrm{UP}^{(e)}$ that are violated by $\vec{p}$ during the phase $e$, and denote those centers as $\{u_1, u_2, \ldots u_s\}$. For each $1 \leq i \leq s$, let $e_i$ be the minimum phase such that $u_i \in U_{end}^{(e')}$ for every $e_i \leq e' \leq e$. We define the arrival order as follows. For any $u_i$ and $u_j$, if $e_i < e_j$, we say $u_i$ arrives earlier than $e_j$. If $e_i = e_j$, let $((C_1 := W^{(e_i)}, \mathrm{wit}_{C_1}), (C_2, \mathrm{wit}_{C_2}) \ldots, (C_{s-1}, \mathrm{wit}_{C_{s-1}}), (C_s := U_{end}^{(e_i)}, \mathrm{wit}_{C_s}))$ be the intermediate center set sequence during the execution of $\mathrm{MAKEROBUST}(\vec{P}^{(e_i)}, (W^{(e_i)}, \mathrm{wit}_{W^{(e_i)}}))$ (which is also $\mathrm{MAKEROBUST}(\vec{P}^{(e_j)}, (W^{(e_i)}, \mathrm{wit}_{W^{(e_j)}})))$. Let $c_i = \min\{t : u_i \in C_t\}$ and $c_j = \min\{t : u_j \in C_t\}$. By the algorithm $\mathrm{MAKEROBUST}$, $c_i \neq c_j$. If $c_i < c_j$ we say $u_i$ arrives earlier than $u_j$. Without loss of generality we write the centers as $(u_1, u_2, \ldots, u_s)$ such that $u_i$ arrives earlier than $u_j$ for any $i \leq j$.

For each $1 \leq i \leq s$, suppose $(u_i, v(u_i))$ is the output of $\mathrm{ROBUSTIFY}$ for a center $u_i' \in U_0^{(e_i)}$. Let $C$ be the center set immediately before calling $\mathrm{ROBUSTIFY}$ to $u_i'$. Let $t(u_i)$ be the smallest positive integer such that $(1+\epsilon)^{\frac{t(u_i)}{z}} \geq \mathrm{dist}(u_i', C \setminus \{u_i'\})/5$. Then by line 1 of Algorithm 2 $v(u_i)$ is $t(u_i)$-prefix robust with respect to $\vec{P}_{end}^{(e_i)}$. Moreover, by the order of the sequence, $u_{i+1}$ must also be in $C$. Because of the minimum property of $t(u_i)$ we have $\mathrm{dist}(u_i', u_{i+1}) \geq 5(1+\epsilon)^{\frac{t(u_i)-1}{z}}$. On the other hand, by Lemma B.3 we have $\mathrm{dist}(u_i', u_i) \leq \frac{\epsilon}{3z}(1+\epsilon)^{\frac{t(u_i)}{z}}$.

Combining these together,

$$\mathrm{dist}(u_i, u_{i+1}) \geq \mathrm{dist}(u_i', u_{i+1}) - \mathrm{dist}(u_i', u_i) \geq \frac{5}{(1+\epsilon)^{\frac{1}{z}}}(1+\epsilon)^{\frac{t(u_i)}{z}} - \frac{\epsilon}{3z}(1+\epsilon)^{\frac{t(u_i)}{z}} \geq \frac{5-\epsilon}{(1+\epsilon)^{\frac{1}{z}}}(1+\epsilon)^{\frac{t(u_i)}{z}}.$$

As $\vec{p}$ violates $u_i$ for every $1 \leq i \leq s$, we have

$$\mathrm{dist}(p, u_i) \leq (1 + \frac{\epsilon}{3z})(1+\epsilon)^{\frac{t(u_i)}{z}}, \forall 1 \leq i \leq s.$$

So it follows that for all $0 \leq i \leq s - 1$,

$$(1+\epsilon)^{\frac{t(u_{i+1})}{z}} \geq \frac{\mathrm{dist}(p, u_{i+1})}{1+\frac{\epsilon}{3z}} \geq \frac{\mathrm{dist}(u_i, u_{i+1}) - \mathrm{dist}(p, u_i)}{1+\frac{\epsilon}{3z}} \geq \frac{(\frac{5-\epsilon}{(1+\epsilon)^{\frac{1}{z}}}) - (\frac{1}{1+\frac{\epsilon}{3z}})}{1+\frac{\epsilon}{3z}}(1+\epsilon)^{\frac{t(u_i)}{z}} \geq 2(1+\epsilon)^{\frac{t(u_i)}{z}}.$$

So the sequence $((1+\epsilon)^{\frac{t(u_i)}{z}})_{1 \leq i \leq s}$ grows exponentially. Thus $s$ can only be at most $O(\log(\Delta))$. This finishes the proof of Lemma C.9. $\square$

Since there are at most $m$ insertions we immediately have the following corollary.

**Corollary C.10.** *There are at most $O(m \log(\Delta))$ centers that are violated by points insertion.*

Now we deal with the second case: centers that are violated by centers update.

**Lemma C.11.** *There are at most $O(\frac{\alpha 2^{O(z \log(z))} m \log^2(\Delta)}{\epsilon^{8z-2}})$ centers that are violated by centers update.*

*Proof.* We construct a collection of queues and a mapping $\phi$ from the center points to those queues to prove the lemma. The elements of every queue in the collection belong to the candidate center set. And the collection is constructed as follows. Recall that a center is in $F(e)$ if it is either in $U_{end}^{(e)} \cap (W^{(e)} \setminus U_0^{(e)})$ or is obtained by calling $\mathrm{ROBUSTIFY}$ to centers in $W^{(e)} \setminus U_0^{(e)}$. For every fresh center $c \in F^{(e)}$, we construct a new queue instance $Q := (c)$ with a single element $c$, labeling the element "fresh center". Moreover, we assign the center $c$ to the queue: $\phi(c) := Q$. Given a phase $e$, if a center $c \in U_0^{(e)}$ is replaced by a center $c'$ by calling $\mathrm{ROBUSTIFY}(c)$ and $c$, suppose $\phi(c) = Q$, we add an element $c'$ in $Q$. Then we let $\phi(c') = Q$ and label the element $c$ by either "violated by points insertion" or "violated by centers update" based on which case $c$ belongs to.

We first show that this procedure is well-defined. Specifically, we show that for every phase $e$, each center in $U_0^{(e)}$ is assigned to a unique queue. This is done by induction on the phase $e$. At the beginning, all vertices are considered as the fresh centers. So each center is assigned to a unique queue. For a phase $e$, suppose every center point $c \in U_0^{(e)}$ is assigned to a unique queue. Consider the next phase $e + 1$ and centers in $U_0^{(e+1)}$. If, $c \in F^{(e)}$. Then the center will be assigned to a new queue thus it is unique. Otherwise, if $c \in \mathrm{UP}^{(e)}$, either $c \in U_0^{(e)}$ or $c \in U_0^{(e+1)} \setminus U_0^{(e)}$. For the first case such that $c \in U_0^{(e)}$, it is assigned to a unique queue by the induction hypothesis. For the second case, there must be a center $c' \in U_0^{(e)}$ such that $c = \textsc{Robustify}(\vec{P}^{(e)}, W^{(e)}, c')$. In this case, by the induction hypothesis, $c'$ is assigned to a unique queue.

Based on the construction, for each queue in this collection, the first element is labeled by "fresh center" and the labels of other elements are consecutive "violated by centers update" and "violated by points insertion" alternatively. We claim that for any queue in the collection, there is no sub-queue with a length more than $O(\frac{z \log(\Delta)}{\epsilon})$ such that every element is labeled by "violated by centers update". This is because every time a center is violated by centers update, by the definition, it means the witness sequence is too short. And the length of the witness after calling $\textsc{Robustify}$ increases by at least 1. This implies $(1+\epsilon)^{\frac{t}{z}}$ increases exponentially where $t$ denotes the length of the witness.

Secondly, Lemma C.7 implies that there are at most $O(\frac{\alpha 2^{O(z \log(z))} m}{\epsilon^{8z-3}})$ elements labeled by "fresh centers" and Lemma C.9 implies that there are at most $O(m \log(\Delta))$ elements labeled by "violated by points insertion" among all queues in the collection. So in the collection, there are most $O(\frac{\alpha 2^{O(z \log(z))} m}{\epsilon^{8z-3}} + m \log(\Delta)) \cdot O(\frac{z \log(\Delta)}{\epsilon}) = O(\frac{\alpha 2^{O(z \log(z))} m \log^2(\Delta)}{\epsilon^{8z-2}})$ elements labeled by "violated by centers update". Thus we finish the proof of Lemma C.11. $\qquad \square$

Finally we combine everything to conclude the proof of Lemma C.6.

*Proof of Lemma C.6.* Combining Lemma C.7, Lemma C.9 and Lemma C.11 shows that

$$
\begin{aligned}
\sum_e \left[ \sum_{1 < i \le \ell_e} |U_i^{(e)} \setminus U_{i-1}^{(e)}| + |U_{end}^{(e)} \setminus U_{\ell_e}^{(e)}| \right] &\le \sum_e \left[ \ell_e + |U_{end}^{(e)} \setminus U_0^{(e)}| + \ell_e \right] \\
&\le O(m) + \sum_e \left[ |F^{(e)}| + |\mathrm{UP}^{(e)} \setminus U_0^{(e)}| \right] \\
&\le O(m) + O(\frac{\alpha 2^{O(z \log(z))} m}{\epsilon^{8z-3}}) + O(m \log(\Delta)) + O(\frac{\alpha 2^{O(z \log(z))} m \log^2(\Delta)}{\epsilon^{8z-2}}) \\
&\qquad\qquad\qquad \text{(by Lemma C.7, Corollary C.10 and Lemma C.11)} \\
&= O(\frac{\alpha 2^{O(z \log(z))} m \log^2(\Delta)}{\epsilon^{8z-2}}).
\end{aligned}
$$

$\qquad \square$

### C.3.1 THE NUMBER OF WELL SEPARATED PAIRS IS LOWER BOUNDED: PROOF OF LEMMA C.8

In this section we prove that the number of well separated pairs between the center set maintained by the algorithm at step 2 and an $\alpha$-approximate solution at the end of the phase is $k - \Omega(\ell_0)$, where $\ell_0$ is the number of centers the algorithm deletes. Our key lemma Lemma C.13 shows that given a point set $P$ and two center sets $U$ and $V$, suppose $U$ and $V$ form $t$ $\epsilon$-well separated pairs, then $\Omega(k - t)$ centers can be removed from $U$ to form $U'$, such that $\mathrm{cost}(P, U') \le \mathrm{cost}(P, U) + O(\epsilon)(\mathrm{cost}(P, U) + \mathrm{cost}(P, V))$. This is an improvement compared with Fichtenberger et al. (2021) where they only provide an upperbound $\mathrm{cost}(P, U') \le \mathrm{cost}(P, U) + O(1)(\mathrm{cost}(P, U) + \mathrm{cost}(P, V))$. We first provide the cost increase bound for re-assigning points to other centers in Lemma C.12. Base on that, we show how to pick a subset with size large enough to delete while the cost is upper bounded in Lemma C.13. Combining the algorithm and Lemma C.13 we conclude our goal in Lemma C.8.

**Lemma C.12.** *Suppose $U$ and $V$ are two center sets, $|U| = |V| = k$, and $\vec{P}$ is a point set. Also suppose $t$ is some integer such that $U$ and $V$ form $t$ $\epsilon$-well separated pairs. Then there exists a subset $D \subseteq U$ with $|D| \geq (k - t)/2$ and a mapping $\phi : D \to U$, such that*

*1. $\forall u \in D, \phi(u) \neq u$*
*2. $\forall p \in P$, suppose $u$ is the nearest neighbor of $p$ in $U$ and $v$ is the nearest neighbor of $p$ in $V$, then*

$$\mathrm{dist}^z(p, \phi(u)) - \mathrm{dist}^z(p, u) \leq \frac{4 \cdot (10z)^{z-1}}{\epsilon^{2z-1}} (\mathrm{dist}^z(p, u) + \mathrm{dist}^z(p, v)).$$

*Proof.* Construction of $D$ and $\phi$: For a center $u \in U$ that is not in a well separated pair, we consider the following case:

Let $v$ be the point in $V$ that is the closest point to $u$.

1. If the cloesest point to $v$ from $U$ is not $u$, we denote it as $u'$. Then we include $u \in D$ and let $\phi(u) := u'$.
2. Otherwise, the cloesest point to $v$ from $U$ is $u$. Let $u'$ be the second closest point to $v$ in $U$. And if it satisfies $\mathrm{dist}(u, v) \geq \epsilon \, \mathrm{dist}(u, u')$, then include $u \in D$ and let $\phi(u) := u'$.
3. If the above does not hold, then the closest point to $v$ from $U$ is $u$ and $\mathrm{dist}(u, v) \leq \epsilon \, \mathrm{dist}(u, u')$. Let $v'$ be the closest point to $v$ from $V$. Then $\mathrm{dist}(u, v) \geq \epsilon \, \mathrm{dist}(v, v')$. If there is a $u'' \in U$ such that $\mathrm{dist}(u'', v') \leq \frac{2}{\epsilon} \, \mathrm{dist}(u, v)$, then include $u \in D$ and let $\phi(u) := u''$.

Next, we show that $|D| \geq (k - t)/2$. We construct a bipartite graph $H$, whose vertices consist of points in $U \cup V$. For all points $u \in U$, let $v$ be the point in $V$ that is closest to $u$. And we put an edge pointing to $v$ from $u$. A similar edges construction is done for $v \in V$.

Let $\widehat{U}$ be the center set containing centers that do not form a well separated pair with any center in $V$. We first prove for each $u$ in $\widehat{U} \setminus D$, the in-degree is at least 2 and the out-degree is 1. For any $u \in \widehat{U} \setminus D$, suppose $u$ points to $v$, and $v' \in V$ is the closest point to $v$. By our selection of $u$, we have that $v$ points to $u$ and for all $u' \in U \setminus \{u\}$, $\mathrm{dist}(u', v') \geq \frac{2}{\epsilon} \, \mathrm{dist}(u, v)$. On the other hand, $\mathrm{dist}(v', v) \leq \frac{1}{\epsilon} \, \mathrm{dist}(u, v)$. So $\mathrm{dist}(u', v) \geq \mathrm{dist}(u', v') - \mathrm{dist}(v', v) \geq \frac{1}{\epsilon} \, \mathrm{dist}(u, v) \geq \mathrm{dist}(u, v)$. So $v'$ also points to $u$. Suppose $|\widehat{U} \setminus D| = n$. Let $Q$ be the subset of $V$ such that each of its vertex points to some $u \in \hat{U} \setminus D$. Then we have $|Q| \geq 2n$.

Consider the points set $D$. The out-degree of the set is $k - t - n$. We claim that there are at least $n$ points in $D$ such that one of the following is true:

1. $u$ points to some $v$ in $Q$.
2. $u$ points to some $v$ in $V \setminus Q$. And $v$ does not point to this $u$.

Suppose in $D$, there are $m$ points pointing to some $v$ in $Q$. If $m \geq n$ then we are done. So we assume $m < n$. Then the number of points in $D$ that do not point to some $v$ in $Q$ is $k - t - n - m$. However, $\hat{V} \setminus Q$ has only at most $k - t - 2n$ points. There are at most $k - t - 2n$ points $u$ pointing to $v \in \hat{V} \setminus Q$ and $v$ pointing to $u$, so there will be at least $(k - t - n - m) - (k - t - 2n) = n - m$ points in $D$ satisfying the second condition. Those $u$'s satisfy the first condition of points in $D$. So we have found at least $n$ points in $D$. This shows $k - t = |D| + |\hat{U} \setminus D| \leq 2|D|$ therefore $|D| \geq (k - t)/2$.

Finally, we prove that the cost increase is bounded. For any $u \in D$, consider the following case:

1. If this $u$ satisfies the first condition. Then $\forall p \in \vec{P}[U, u]$, let $v_0 \in V$ that $p$ is assigned to. We have

$$\text{dist}^z(p, \phi(u)) - \text{dist}^z(p, u) \le \epsilon \, \text{dist}^z(p, u) + (\frac{\epsilon + 2z}{\epsilon})^{z-1} \text{dist}^z(u, \phi(u))$$

$$\le \epsilon \, \text{dist}^z(p, u) + (\frac{2(\epsilon + 2z)}{\epsilon})^{z-1}[\text{dist}^z(u, v) + \text{dist}^z(v, \phi(u))]$$

$$\le \epsilon \, \text{dist}^z(p, u) + 2(\frac{2(\epsilon + 2z)}{\epsilon})^{z-1} \text{dist}^z(u, v)$$

$$\le \epsilon \, \text{dist}^z(p, u) + 2(\frac{2(\epsilon + 2z)}{\epsilon})^{z-1} \text{dist}^z(u, v_0)$$

$$\le \epsilon \, \text{dist}^z(p, u) + 2(\frac{4(\epsilon + 2z)}{\epsilon})^{z-1}[\text{dist}^z(p, u) + \text{dist}^z(p, v_0)]$$

$$\le 3(\frac{10z}{\epsilon})^{z-1}[\text{dist}^z(p, u) + \text{dist}^z(p, v_0)].$$

2. If this $u$ satisfies the second condition. Then $\forall p \in \vec{P}[U, u]$, let $v_0 \in V$ that $p$ is assigned to. We have

$$\text{dist}^z(p, \phi(u)) - \text{dist}^z(p, u) \le \epsilon \, \text{dist}^z(p, u) + (\frac{\epsilon + 2z}{\epsilon})^{z-1} \text{dist}^z(u, \phi(u))$$

$$\le \epsilon \, \text{dist}^z(p, u) + (\frac{\epsilon + 2z}{\epsilon})^{z-1} \frac{1}{\epsilon^z} \text{dist}^z(u, v)$$

$$\le \epsilon \, \text{dist}^z(p, u) + (\frac{\epsilon + 2z}{\epsilon})^{z-1} \frac{1}{\epsilon^z} \text{dist}^z(u, v_0)$$

$$\le \epsilon \, \text{dist}^z(p, u) + (\frac{2(\epsilon + 2z)}{\epsilon})^{z-1} \frac{1}{\epsilon^z} (\text{dist}^z(p, u) + \text{dist}^z(p, v_0))$$

$$\le \frac{2 \cdot (5z)^{z-1}}{\epsilon^{2z-1}}[\text{dist}^z(p, u) + \text{dist}^z(p, v_0)]$$

3. If this $u$ satisfies the third condition. Then $\forall p \in \vec{P}[U, u]$, let $v_0 \in V$ that $p$ is assigned to. We have

$$\text{dist}^z(p, \phi(u)) - \text{dist}^z(p, u) \le \epsilon \, \text{dist}^z(p, u) + (\frac{\epsilon + 2z}{\epsilon})^{z-1} \text{dist}^z(u, \phi(u))$$

$$\le \epsilon \, \text{dist}^z(p, u) + (\frac{(\epsilon + 2z)}{\epsilon})^{z-1}[4^{z-1} \text{dist}^z(u, v) + 4^{z-1} \text{dist}^z(v, v') + 2^{z-1} \text{dist}^z(v', \phi(u))]$$

$$\le \epsilon \, \text{dist}^z(p, u) + (\frac{4(\epsilon + 2z)}{\epsilon})^{z-1}(1 + \frac{2}{\epsilon^z}) \text{dist}^z(u, v)$$

$$\le \epsilon \, \text{dist}^z(p, u) + (\frac{4(\epsilon + 2z)}{\epsilon})^{z-1}(1 + \frac{2}{\epsilon^z}) \text{dist}^z(u, v_0)$$

$$\le \frac{4 \cdot (10z)^{z-1}}{\epsilon^{2z-1}}[\text{dist}^z(p, u) + \text{dist}^z(p, v_0)].$$

$\square$

**Lemma C.13.** *Let $\vec{P}$ be a weighted point set, $\epsilon \in (0, 1)$ and $U$ and $V$ be two center sets. Let $t$ be some integer such that $U$ and $V$ have $t$ $\epsilon$-well separated pairs. Then for any $\alpha < \frac{(10z)^{z-1}}{\epsilon^{2z-1}}$, there exists a subset $S \subseteq U$ with size at least $\frac{\alpha \epsilon^{2z-1}(k-t)}{6(10z)^{z-1}}$ such that*

$$\text{cost}_z(\vec{P}, U \setminus S) \le \text{cost}_z(\vec{P}, U) + 4\alpha(\text{cost}_z(\vec{P}, U) + \text{cost}_z(\vec{P}, V)). \tag{5}$$

*Proof.* First we apply Lemma C.12. It says that we can find a set $D \subseteq U$ with size $\frac{k-t}{2}$ containing only those points that do not form a $\epsilon$-well separated pair. There exists a mapping $\phi : D \to U$ such that reassigning the cluster in $D$ to $\phi(D)$ will lead to the cost increment less than $\frac{4 \cdot (10z)^{z-1}}{\epsilon^{2z-1}}(\text{cost}_z(\vec{P}, U) + \text{cost}_z(\vec{P}, V))$.

Consider the abstract graph $H$ where the nodes are the elements of $U$ and there is a directed arc from $D$ to $\phi(D)$. More formally, $H = (U, \{(u, \phi(u)) \mid u \in D\})$. Notice that every node of $H$ has

outdegree at most 1. Thus, there exists a coloring of the nodes of $H$ with three colors, such that all arcs are dichromatic. Let $\hat{S}$ denote the color set with the largest number of nodes of $D$. We have that $\hat{S}$ contains at least $|D|/3$ nodes of $D$.

By $\alpha < \frac{(10z)^{z-1}}{\epsilon^{2z-1}}$, we can arbitrarily partition $\hat{S}$ into $\frac{(10z)^{z-1}}{\alpha\epsilon^{2z-1}}$ parts, each of cardinality at least $\frac{\alpha\epsilon^{2z-1}|D|}{3(10z)^{z-1}}$. Given a vertex $u$ with outdegree 1, we define the reassignment operation as follows. For all $\vec{p} \in \vec{P}_U[u]$, let $\phi(u)$ be the center serving it. Note that deleting $\hat{S}$ increases cost at most $\frac{4\cdot(10z)^{z-1}}{\epsilon^{2z-1}}(\mathrm{cost}_z(P, \vec{U}) + \mathrm{cost}_z(P, \vec{V}))$. By an averaging argument, there exists a set such that reassigning each cluster in this set increases the cost by at most $4\alpha(\mathrm{cost}_z(\vec{P}, U) + \mathrm{cost}_z(\vec{P}, V))$. Denote this set as $S$ and it is what we want to obtain. Thus deleting the centers in $S$ will increase the cost by at most $4\alpha(\mathrm{cost}_z(\vec{P}, U) + \mathrm{cost}_z(\vec{P}, V))$. Since the arcs of $H$ are dichromatic, if $u \in S$ then $\phi(u) \notin S$. So the reassignment is well-defined.

$\qquad\square$

Now we fix a phase $e$ in our algorithm. Let $V$ be an $\alpha$-approximate solution at the end of the phase $e$. We show that if one can delete at most $\ell$ centers from $U_0^{(e)}$ such that the cost increases by a factor of $(1 + 12\epsilon)\alpha$, then the number of well separated pairs between sets $U_0^{(e)}$ and $V$ is lower bounded.

*Proof of Lemma C.8.* Let $t$ be the number of $\frac{\epsilon^4}{200z}$-well separated pairs between $U_0$ and $V$. By letting $\alpha = \frac{\epsilon}{\alpha}$ in Lemma C.13, we know that there is a set $S$ with $|S| \geq \bar{\ell} := \Omega(\frac{\epsilon^{8z-3}(k-t)}{\alpha 2^{O(z\log(z))}})$, and $\mathrm{cost}_z(\vec{P}_0, U_0 \setminus S) \leq \mathrm{cost}_z(\vec{P}_0, U_0) + \frac{4\epsilon}{\alpha}(\mathrm{cost}_z(\vec{P}_0, U_0) + \mathrm{cost}_z(\vec{P}_0, V))$. This is upper bounded by $\mathrm{cost}_z(\vec{P}_0, U_0) + 12\epsilon\,\mathrm{cost}_z(\vec{P}_0, U_0)$, because $\mathrm{cost}_z(\vec{P}_0, V) \leq \mathrm{cost}_z(\vec{P}_{\ell+1}, V) \leq \alpha\,\mathrm{OPT}_z(\vec{P}_{\ell+1}) \leq 2\alpha\,\mathrm{cost}_z(\vec{P}_0, U_0)$. Furthermore, there exists a set $S'$ with $|S'| \geq |S|/2 = \bar{\ell}/2$ such that the cost after deleting $S'$ from $U_0$ is upper bounded by $\mathrm{cost}_z(\vec{P}_0, U_0) + 6\epsilon\,\mathrm{cost}_z(\vec{P}_0, U_0)$. Therefore, if the minimal cost after deleting $\ell$ center for some $\ell < k$ is larger than $\mathrm{cost}_z(\vec{P}_0, U_0) + 6\epsilon\,\mathrm{cost}_z(\vec{P}_0, U_0)$, we must have $\ell \geq \bar{\ell}/2$.

For our algorithm, if $\ell = 0$ then it does not delete any center. In this case we must have $\bar{\ell} \leq 1$. So $t \geq k - \Omega(1)$. From now on we assume $\ell > 0$. For every $i < k$, suppose the algorithm outputs the center set $D_i \subseteq U_0$ with $|D_i| = k - i$. If there is some $i < k$ such that

$$\alpha(1 + 6\epsilon)\,\mathrm{cost}_z(\vec{P}_0, U_0) \leq \mathrm{cost}_z(\vec{P}_0, U_0 \setminus D_i) \leq \alpha(1 + 12\epsilon)\,\mathrm{cost}_z(\vec{P}_0, U).$$

Denote $\mathrm{OPT}^{(i)}$ as the minimal cost by deleting $i$ centers from $U_0$. Then we know $\mathrm{OPT}^{(i)} \geq (1 + 6\epsilon)\,\mathrm{cost}_z(\vec{P}_0, U_0)$ so we have $i \geq \bar{\ell}/2$.

Otherwise, if for any $i < k$, either $\mathrm{cost}_z(\vec{P}_0, U_0 \setminus D_i) \leq \alpha(1 + 6\epsilon)\,\mathrm{cost}_z(\vec{P}_0, U_0)$, or $\alpha(1 + 12\epsilon)\,\mathrm{cost}_z(\vec{P}_0, U) \leq \mathrm{cost}_z(\vec{P}_0, U_0 \setminus D_i)$. In this case, the algorithm picks the largest $i$ such that the first inequality holds. And this is exactly $\bar{\ell}$.

Based on the above discussion, we always have $\ell \geq \bar{\ell}/2 \geq \Omega(\frac{\epsilon^{8z-3}(k-t)}{\alpha 2^{O(z\log(z))}})$. Thus $t \geq k - O(\alpha 2^{O(z\log(z))}\ell/\epsilon^{8z-3})$. This finishes the proof of Lemma C.8. $\qquad\square$

## D   EXPERIMENT FOR $k = 5$ AND $k = 20$

In this section we give the cost and consistency curves in Figures 3 and 5 for $k = 5$ and $k = 20$. Both "ours-faithful" and "ours-heuristic" have significant lower consistency compared with the baselines, especially for larger $k$ and larger dataset. We also list the accumulative running time for the algorithms. "ours-heuristic" outperforms all other algorithms. As can be seen from the figure, "ours-faithful" has a comparable running time (within 150%) to all baselines, except for the case of COVERTYPE ($k = 20$) where our algorithm may perform no more than 4 times worse; Nonetheless, even for this relatively bad case of COVERTYPE with $k = 20$, "ours-faithful" still runs in about 0.02 seconds per insertion on average, which is fast in an absolute sense. This is a reasonable performance given that our algorithm uses much more involved steps to maintain consistency. Indeed,

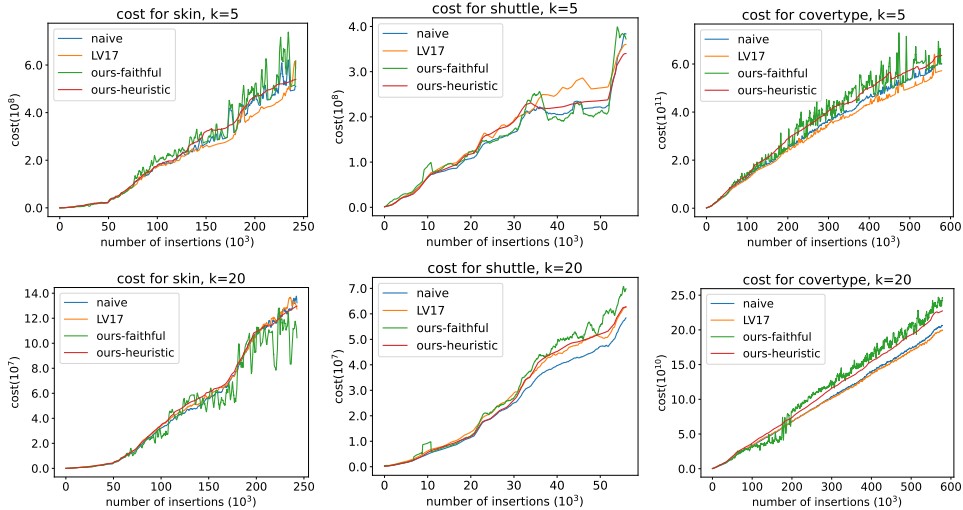

Figure 3: The cost curve over the insertions of points, for all datasets and different choices of $k$. We plot the curve after applying a moving average with a window size equal to 1% of the dataset size.

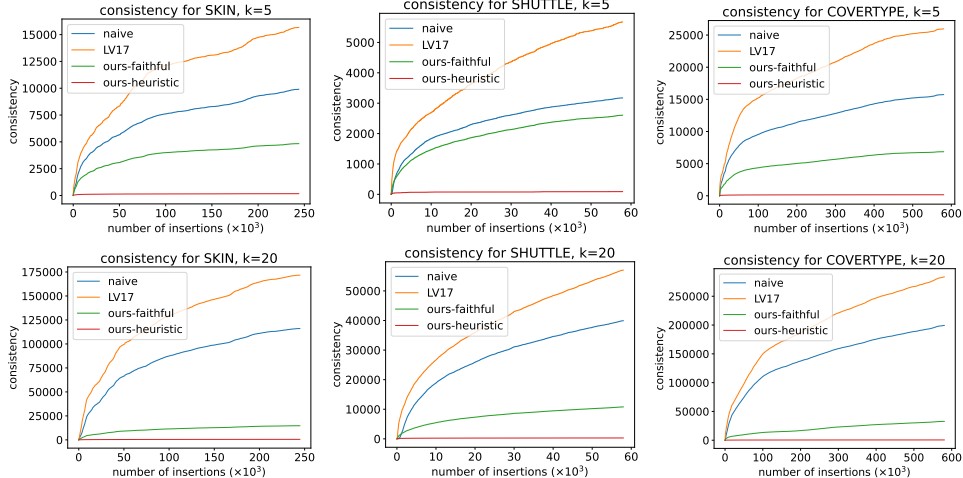

Figure 4: The consistency curve over the insertions of points, for all datasets and different choices of $k$.

the mentioned behavior of the running time also matches our (theoretical) running time bound. To see this, since we plug in $k$-MEANS++, our total running time is $\tilde{O}(nk + k^5)$. However, the "LV17" baseline runs in roughly $\tilde{O}(nk + k^3)$ time. When $k = 20$, the $k^5$ term starts to dominate our running time, and hence our running time curve starts to deviate from that of "LV17".

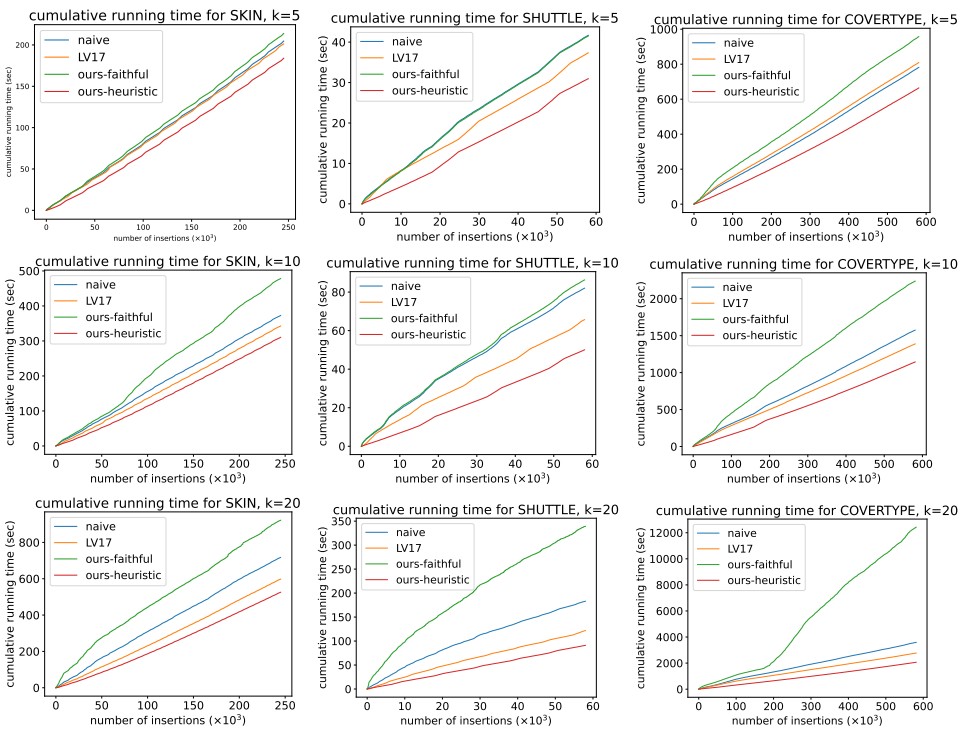

Figure 5: The cumulative running time over the insertions of points, for all datasets and different choices of $k$. $x$-axis corresponds to the number of insertions and $y$-axis corresponds to the cumulated running.

