# OpenReview forum: "Online Clustering with Nearly Optimal Consistency"
_ICLR.cc/2025/Conference — ICLR 2025 Poster_

### Official Review · Reviewer_aeE5 · 2024-11-04

**Soundness:** 3
**Presentation:** 2
**Contribution:** 3
**Rating:** 8
**Confidence:** 3

**Summary:**

The paper gives an algorithm for online $(k,z)$ clustering. The algorithm uses an 'online' coreset and is able to give theoretically best-known consistency bounds for the general $ (k,z)$ clustering problem. For an $\alpha$ approximate offline algorithm for clustering, the paper guarantees a $(1 +\epsilon) \alpha^2$ competitive ratio implying that given an optimal offline algorithm, it is theoretically possible to achieve $(1 + \epsilon) $ competitive ratio. Empirical evaluations on 3 datasets and comparison with 2 baselines, one of them being the popular $k$-means++ algorithm show the effectiveness of the algorithm in giving superior consistency.

**Strengths:**

1) Online clustering is a practically more important problem and the idea of 'consistency', which loosely tries to address the problem with not allowing an algorithm to change decision in an online setting, has gained significant attention in the recent past. As such the paper will be of considerable interest and significance.
2) The paper generalizes and improves the theoretical results for online clustering problem. The paper does a good job of describing the high-level algorithm and also the technical challenges faced in extending as well as improving the results. Although many ideas/definitions are borrowed from existing works the paper does a good job of combining these to address the challenges. I could not go through all the proofs, but the paper overall appears technically solid.
3) Experimental results show the algorithm achieves much better 'consistency' than the baselines while incurring comparable costs.

**Weaknesses:**

1) The paper is not easy to follow. The notations are dense. Also, it requires some familiarity with existing work to understand the paper as it relies heavily on ideas from the related work. Moreover, there are many typos throughout the paper.
For e.g.:
Line 86 bonded instead of bounded
Line 144 and 145 approximation and approximate are spelt incorrectly

2) Experiments could have used other baselines for online k -means clustering. Also, what is the time required empirically to make updates at every step of the online algorithm could be included.

**Questions:**

1) In the coreset definition 2.4 it is given that $\overrightarrow{S} \subseteq \overrightarrow{P}$. However, points in a coreset can have higher weights than original data points. Am I missing something? Please clarify.

2) In the experiments in your method, is the cost calculated over the coreset, or centers are selected using coreset and then cost is calculated on original data?

---

### Official Review · Reviewer_d1TJ · 2024-11-05

**Soundness:** 4
**Presentation:** 3
**Contribution:** 3
**Rating:** 8
**Confidence:** 4

**Summary:**

This paper's considers online (k,z) clustering. Here, z is simply a parameter used as an exponent for the distance function (i.e. z=2 for k-means, and z=1 for k-medians). In this setting, on each subsequent point the learner outputs a set of precisely k points. Algorithms are evaluated with respect to a bicriteria, with 1. their approximation factor being the amount they differ from the optimal clustering over the set of points seen thus far, and 2. their consistency being the total amount of changes they make to their centers.

This paper's contribution is an online algorithm that achieves a $(1+\epsilon)\alpha^2$ approximation factor (where $\alpha$ is the approximation factor of any offline clustering algorithm being taken as input), and $O(\alpha d k poly(\epsilon^{-z}))$ consistency. When $\alpha$ equals $1$, this is the first algorithm to achieve $O(k)$ consistency with a $(1+\epsilon)$ approximation factor.

This paper builds on prior work by using a well-known technique of first maintaining a coreset that approximates the clustering behavior of the dataset, and then applying an offline clustering algorithm to the coreset. However, the key technical challenge is in applying the offline clustering algorithm in a manner that does not change the set of outputted cluster centers by too much. The net result of both of these factors is that because the corset changes less frequently than the dataset, and because the offline clustering algorithm is being cleverly applied, the net amount of center change is small which gives us the linear $O(k)$ consistency.

The key technical innovation of this paper lies within its second step, which proposes an online clustering algorithm (over weighted points) that, under certain conditions achieves $O(m)$ consistency (where $m$ is the number of points). Note that this algorithm will be utilized while substituting $m = O(\log(\Delta n))$ ($\Delta$ is the aspect ratio) and this will achieve the desired results outlined above.

The main idea behind their algorithm is to construct a "robust center sequence," for the online set of points. This sequence essentially picks a sequence of points where each point represents an "appropriate" substitution for an initial point based on exponentially increasing distance scales. Then, from the robust center sequence, they show how to appropriately select $k$ centers, along with how to maintain such a sequence.

**Strengths:**

This paper achieves a very exciting overall result -- $(1+\epsilon)$-approximation with $O(k)$-consistency is more or less the best we can reasonably expect for this problem -- all other improvemenets would only be in shaving down constants. Furthermore, the algorithm is reasonably efficient -- it is polynomial in all aspects except for $\epsilon$ (which can always be set to even a reasonably large value and still achieve interesting results).

**Weaknesses:**

I think the paper could use improvement on its presentation. I'd suggest moving the discussion of Lemma 3.3 further up in the paper and ensuring that the reader can easily grasp the intuition and definition behind robust sequences. I also think that it might help to simply choose a value of z for the main body and defer details of the general proof to the appendix.

Finally, I found the constant references to Fichtenberger et. al. in section 4 cumbersome. I understand that many of the main ideas are taken from that paper, but this can simply be addressed in the related works section. The way this section is currently written gives the reader the impression that they should read Fichtenberger first, and I don't think this should be necessary if the intuitions behind the definitions are explained properly.

**Questions:**

Can you confirm (to check my understanding) that main strength of the result shown in lemma 3.3. is that the consistency does not depend on k? If i understood the setting correctly, obtaining a consistency of $O(mk)$ would always be trivial (just pick $k$ new centers on each point).

---

### Official Review · Reviewer_7dtW · 2024-11-09

**Soundness:** 2
**Presentation:** 3
**Contribution:** 2
**Rating:** 5
**Confidence:** 4

**Summary:**

The paper studies the online version of the classic k-means clustering problem. Here, points arrive one after another and at every step, the algorithm must decide whether to create a new center or not. The objective they consider is called "consistency", which informally measures the total number of times an algorithms "changes" a cluster center. (At least k changes are required for any online algorithm.) The main result of the paper is an online algorithm for k-means with consistency $O(k polylog(n))$ (times a factor that depends on the aspect ratio of the dataset).

The authors first give an online algorithm that produces a coreset for the points, while having around k * polylog(n) points. They then use the framework of Fichtenberger et al. (2021) to obtain an algorithm that maintains a k means solution. Most of the technical novelty of the paper is in the part of designing an online algorithm for the "bounded input" case.

**Strengths:**

- k-means clustering is a classic problem in ML and online algorithms for it and other variants of clustering have been an extensive topic of research.
- It is nice that the algorithm nearly matches the lower bounds from the work of Lattanzi and Vassilvitskii.

**Weaknesses:**

- The paper's contributions feel a bit incremental. While there are some novel ideas required in, e.g., Algorithms 2 and 3 and their analysis, the overall framework follows that of prior work on k-median, as well as standard ideas in the literature.

- The paper misses comparison to some prior work. E.g., see:
https://proceedings.mlr.press/v201/bhattacharjee23b.html
http://proceedings.mlr.press/v132/moshkovitz21a/moshkovitz21a.pdf
https://proceedings.mlr.press/v117/bhaskara20a.html

As far as I can tell, these works do not study recourse/consistency explicitly, but they have very similar reasons for requiring to depend on \log \Delta, etc.

- The experiments do not seem to add much value to the paper (and seem disconnected); are there simplifications that are made there that perhaps would make the algorithms easier to implement in practice?

**Questions:**

If the authors can address the weaknesses described above (motivate the novelty of the results a bit more) it would help the paper.

---

### Meta-Review · Area_Chair_MQtD · 2024-12-26

**Metareview:**

The paper considers the problem of online $k$-means clustering (and slightly more generally $(k, z)$-clustering. In order to achieve non-trivial results (as is shown to be necessary in prior work), the authors allow the online algorithm to change the centers, but keep track of the recourse cost, called consistency, which is the total number of center changes across the entire online sequence. The present paper shows that an offline algorithm with $\alpha$-optimality can be converted to an online algorithm with $(1 + \epsilon) \alpha^2$ competitive algorithm with consistency bounded by $O(k \mathrm{polylog}(n))$ (this also requires polynomial in $n$ bounded aspect ratio). This makes significant progress over prior work and almost matches the lower bound in the Lattanzi an Vassilvitskii paper which introduced the framework. In terms of techniques, the paper relies on coresets, which is standard in this area, but there are some innovations required to make everything work.

**Additional Comments On Reviewer Discussion:**

There was robust discussion with the reviewers. Ultimately, I sided with the positive reviewers as this does essentially close an open problem that appeared in an ICML 2017 paper and it does require new ideas.

---

### Decision · Program_Chairs · 2025-01-22

Accept (Poster)